# The secreted protease Adamts18 links hormone action to activation of the mammary stem cell niche

Dalya Ataca [1,6], Patrick Aouad[1,6], Céline Constantin[1,6], Csaba Laszlo[1], Manfred Beleut [1,4], Marie Shamseddin [1,2], Renuga Devi Rajaram[1], Rachel Jeitziner[1,5], Timothy J. Mead [3], Marian Caikovski[1,5], Philipp Bucher[1], Giovanna Ambrosini[1], Suneel S. Apte[3] & Cathrin Brisken [1✉]

Estrogens and progesterone control breast development and carcinogenesis via their cognate receptors expressed in a subset of luminal cells in the mammary epithelium. How they control the extracellular matrix, important to breast physiology and tumorigenesis, remains unclear. Here we report that both hormones induce the secreted protease Adamts18 in myoepithelial cells by controlling *Wnt4* expression with consequent paracrine canonical Wnt signaling activation. *Adamts18* is required for stem cell activation, has multiple binding partners in the basement membrane and interacts genetically with the basal membrane-specific proteoglycan, *Col18a1*, pointing to the basement membrane as part of the stem cell niche. In vitro, ADAMTS18 cleaves fibronectin; in vivo, *Adamts18* deletion causes increased collagen deposition during puberty, which results in impaired Hippo signaling and reduced *Fgfr2* expression both of which control stem cell function. Thus, Adamts18 links luminal hormone receptor signaling to basement membrane remodeling and stem cell activation.

[1] Ecole Polytechnique Fédérale de Lausanne, Station 19, CH-1015 Lausanne, Switzerland. [2] Wellcome Sanger Institute, Wellcome Genome Campus, Hinxton, Cambridge CB10 1SA, UK. [3] Department of Biomedical Engineering-ND20, Cleveland Clinic Lerner Research Institute, 9500 Euclid Ave., Cleveland, OH 44195, USA. [4] Present address: Medoderm GmbH, Robert Koch-Straße 50 D, 55129 Mainz, Germany. [5] Present address: Swiss Institute of Bioinformatics, Agora Swiss Cancer Center Leman, Rue du Bugnon 25a, 1015 Lausanne, Switzerland. [6] These authors contributed equally: Dalya Ataca, Patrick Aouad, Céline Constantin. ✉email: cathrin.brisken@epfl.ch

The breast is the only organ to develop mostly after birth. Milk ducts arborize from the nipple and grow into a specialized subcutaneous stroma called the mammary fat pad in mice. The ductal wall comprises a bi-layered epithelium with the inner luminal cells and outer myoepithelial cells. The epithelium is separated from the stroma by specialized extracellular matrix (ECM), the basement membrane (BM). The ovarian hormones, estrogens and progesterone, are key drivers of mammary gland development and also influence breast carcinogenesis[1]. Both estrogen receptor α (ER) and progesterone receptor (PR) are members of the nuclear receptor family and are readily detected by immunohistochemistry (IHC) in a subset of luminal cells[1]. Activation of hormone receptor signaling in cells with high hormone receptor expression, termed sensor cells[2], triggers the expression of paracrine factors such as amphiregulin and Rankl, which are required for mammary epithelial cell proliferation[3,4] as well as Wnt4 and Cxcl12, which activate stem/progenitor cells[5,6]. Mammary stem and progenitor cells have been identified and characterized based on cell surface markers and functional assays[7]. However, the precise cellular and biochemical components of the stem cell niche and its endocrine regulation remain poorly defined.

Evidence has been provided that mammary ECM can reprogram non-mammary cells to form mammary glands[8,9], suggesting that it contains critical cues for epithelial development. Hedgehog signaling acts via Gli2 downstream of growth hormone receptor signaling in fibroblasts to trigger changes in paracrine signaling and ECM proteins that affect stem cell function[10]. This suggests that stromal fibroblasts are part of the niche under direct endocrine control by growth hormone. Stromal changes accompany different morphogenic processes induced by epithelial hormone signaling and are a hallmark of breast carcinogenesis. Indeed, high radiographic density, which reflects an increase in fibrillar collagen content in the breast stroma, is the single most important risk factor for breast cancer and correlates with progesterone exposure[11,12]. How ECM and stroma are controlled by the major endocrine drivers of breast development and carcinogenesis, epithelial ER and PR signaling, remains elusive.

ADAMTS18 is an orphan member of the A Disintegrin-like And Metalloproteinase domain with ThromboSpondin type 1 Motifs (ADAMTS) family of secreted Zn-dependent metalloproteinases[13] that comprises 19 members[14,15]. Like other zinc metalloproteinases, ADAMTS catalytic activity depends on zinc ion binding within the active site; unique to ADAMTSs is an ancillary domain containing thrombospondin type 1 repeats[16]. ADAMTS proteases are synthesized as precursors with an N-terminal propeptide, which is excised by pro-protein convertases such as furin[14]. Some ADAMTSs process ECM components such as fibrillar collagens, while others are implicated in turnover of the chondroitin sulfate proteoglycans aggrecan and versican[14], and ADAMTS13 uniquely cleaves von-Willebrand factor to maturity[17]. We have previously reported that Adamts18 is required for eye, lung and female reproductive tract and kidney development in the mouse[18]. It is highly homologous to Adamts16, which has a role in renal development and fertility[19,20] and can cleave fibronectin[21]. Here, we show that Adamts18 provides a mechanistic link between epithelial steroid hormone receptor signaling and changes in the ECM, in particular the BM, that regulate mammary epithelial stemness.

## Results

***Adamts18* expression is driven by the *PR/Wnt4* axis.** To elucidate the mechanisms, by which PR signaling in luminal mammary epithelial cells may elicit ECM changes, we sought genes induced in vivo by progesterone treatment[22,23] that fulfilled two criteria: (1) They encoded secretory proteins and (2) They showed delayed induction by progesterone as expected of any indirect PR target which is expressed by myoepithelial cells and can hence directly interact with the BM. *Adamts18* induction was detected at 16 hours (h) and 78 h but not at 4 h[22] and at 24 h but not 8 h following progesterone stimulation[23]. RT-PCR analysis of fluorescence activated cell sorting (FACS)-sorted cells from adult mammary glands showed a 7-fold enrichment of *Adamts18* mRNA in myoepithelial (Lin⁻ CD24⁺ CD49f⁺) over luminal (Lin⁻ CD24⁺ CD49f⁻) cells (Fig. 1a), in line with recent single cell RNA sequencing data[24,25], confirming expression in myoepithelial cells.

Analysis of *Adamts18* transcript levels at different stages of mammary gland development revealed low prepubertal expression that increased 2.7, 7- and 8.6-fold in 4-, 6- and 8-week-old females, respectively; expression rose further during pregnancy with a peak at mid-pregnancy day10.5/12.5 (Fig. 1b). RNAscope in situ hybridization for *Adamts18* transcripts combined with immunofluorescence (IF) for the myoepithelial marker α-smooth muscle actin (Sma) confirmed myoepithelium-specific expression of *Adamts18* in pubertal and adult mammary ducts (Fig. 1c, d). The increased *Adamts18* expression during pregnancy was not attributable to generalized but rather to myoepithelium-specific upregulation of expression (Fig. 1e). Thus, *Adamts18* expression in the mammary epithelium is developmentally regulated, and its mRNA is enriched in myoepithelial cells, making it an attractive candidate to mediate ECM changes downstream of epithelial hormone action.

Next, we tested whether endocrine factors contribute to developmental *Adamts18* expression. First, we mimicked pubertal estrogen stimulation by injecting ovariectomized 21-day-old mice with 17-β-estradiol. Within 18 h of injection, *Adamts18* transcript levels in extracts from total mammary glands increased 1.76-fold (Fig. 1f). Second, we asked whether changes in progesterone levels as they occur during estrous cycles affect *Adamts18* transcript levels and obtained mammary gland extracts from mice in estrus and diestrus. Progesterone plasma levels determined by liquid chromatography-mass spectrometry were on average 2.8-fold higher in diestrus than in estrus (Fig. 1g); *Adamts18* transcript levels in the mammary glands were 1.6-fold higher in diestrous over estrous (Fig. 1h). Thus, physiological *Adamts18* expression correlates with plasma progesterone levels, suggesting that it is progesterone-responsive. The subtle increases in transcript levels are consistent with myoepithelial cells representing a minor fraction of the mammary cell types and hence of the total RNA in the whole tissue extracts we analyzed.

To determine whether epithelium-intrinsic PR signaling is required for *Adamts18* mRNA expression, mammary epithelia from *WT.EGFP⁺* and *PR⁻/⁻.EGFP⁺* mice were grafted to contralateral fat pads of *WT* recipients surgically cleared of the endogenous epithelium and allowed to grow out for six weeks. At sacrifice, reconstitution was validated by fluorescence stereomicroscopy of the engrafted glands. *Adamts18* transcript levels in the mammary glands successfully reconstituted with *PR⁻/⁻* epithelium were on average 27% of those in the contralateral controls (Fig. 1i). Thus, epithelial *PR* expression is required for *Adamts18* mRNA expression.

*Wnt4* is a plausible candidate to induce *Adamts18* expression in myoepithelial cells because it is a PR target[26] transcribed exclusively in PR+ luminal cells[6] and activates canonical Wnt signaling in the myoepithelial cells[6], which express *Adamts18*. We analyzed expression of various Wnt signaling components expressed in the mammary epithelium by RT-PCR in contralateral glands engrafted with *WT.EGFP⁺* and *PR⁻/⁻.EGFP⁺* mammary epithelia. Among the Wnt genes, only *Wnt4* transcript levels were significantly lower in the mutant grafts, furthermore

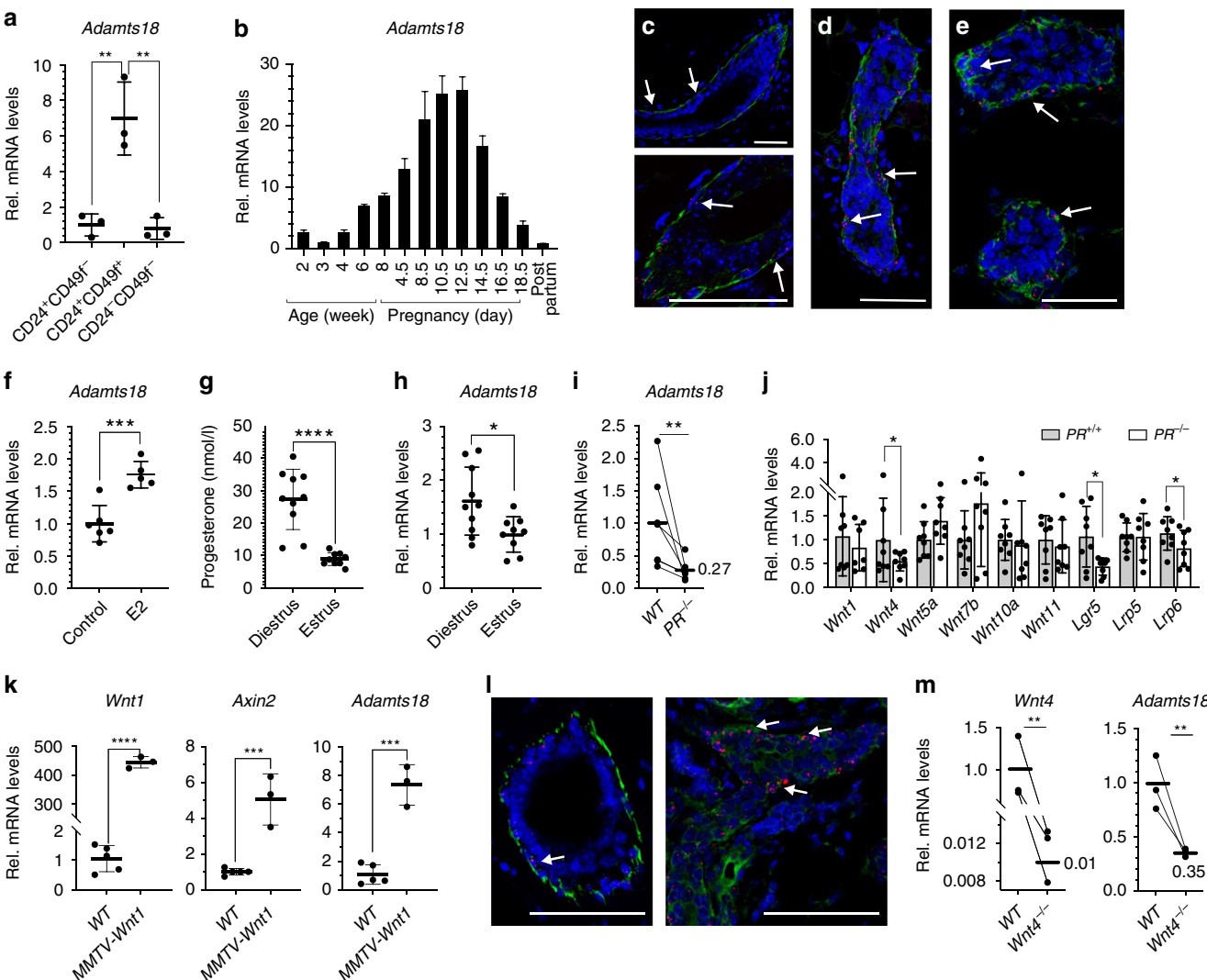

**Fig. 1 _Adamts18_ expression in the mouse mammary gland. a** Dot plot showing _Adamts18_ mRNA expression normalized to _Hprt_ in FACS-sorted CD24+ CD49f− (luminal), CD24+ CD49f+ (myoepithelial) and CD24− CD49f− (stromal) cells. Data represent mean ± SD from $n = 3$ independent experiments. Student _t_-test, two-tailed. **b** Bar plot showing _Adamts18_ mRNA levels normalized to _Hprt_ in mammary glands at different developmental stages. Each bar represents pool of 3 mice, mean ± SD for technical replicates. **c–e** Representative micrographs showing _Adamts18_ mRNA localization in mouse mammary gland during puberty (**c**), adulthood (**d**) and pregnancy day 12.5 (**e**). Red dots represent _Adamts18_ in situ hybridization signal, green: α-Sma, blue: DAPI, arrows show myoepithelial cells; scale bar, 50 µm. **f** Relative _Adamts18_ transcript levels normalized to _Krt5_ in mammary glands from 6 control and 5 E2-treated mice. Data represent mean ± SD, unpaired Student _t_-test, two-tailed. **g** Dot plot showing plasma progesterone levels determined by LC/MS during diestrus ($n = 10$) or estrus ($n = 9$). Data represent mean ± SD, Student _t_-test, two-tailed. **h** Dot plot showing _Adamts18_ mRNA levels normalized to _Krt5_ in mammary glands from mice shown in **g**. Data represent mean ± SD, Student _t_-test, two-tailed. **i** Dot plot showing _Adamts18_ mRNA normalized to _Hprt_ in 6 contralateral mammary glands transplanted with _WT.EGFP_+ or _PR−/−.EGFP_+ epithelium. **j** Bar graph showing relative transcript expression of different Wnt signaling components normalized to _Hprt_ in contralateral glands of 8 mice transplanted with _WT.EGFP_+ and _PR−/−.EGFP_+ epithelia. Each data point represents one gland, mean ± SD, paired Student _t_-test, two-tailed. **k** Dot plots showing relative transcript levels of _Wnt1_, _Axin2_ and _Adamts18_ normalized to _Hprt_ in mammary glands from 5 _WT_ and 3 _MMTV-Wnt1_ virgin mice. Data represent mean ± SD, Student _t_-test, two-tailed. **l** Representative micrographs of _Adamts18_ mRNA localization, (red) dots, in mammary glands from 3 _WT_ and 3 _MMTV-Wnt1_ females, α-Sma (green) and DAPI (blue); arrows show myoepithelial cells. Scale bar, 50 µm. **m** Dot plots showing mRNA levels of _Wnt4_ and _Adamts18_ normalized to _Hprt_ in contralateral glands of 3 mice transplanted with _WT.EGFP_+ and _Wnt4−/−.EGFP_+ epithelia harvested at 8.5-day of pregnancy. *$p < 0.05$; **$p < 0.01$; ***$p < 0.001$; ****$p < 0.0001$.

the transcript levels of the stem cell marker _Lgr5_ and the Wnt co-receptor _Lpr6_ were decreased (Fig. 1j). Consistent with canonical Wnt signaling activation downstream of PR/Wnt4 controlling _Adamts18_ expression, TCF4 binding sites were reported in the _Adamts18_ promoter by ChIP-seq analysis[27]. To assess whether canonical Wnt signaling controls _Adamts18_ expression in vivo, we analyzed _Adamts18_ expression in mammary glands with hyperactive canonical Wnt signaling in the myoepithelium[6] due to the presence of an _MMTV-Wnt1_ transgene[28]. Ectopic

_Wnt1_ expression was readily detected in transgenic glands and expression of the canonical Wnt signaling target, _Axin2_, was increased 5-fold over the non-transgenic control while _Adamts18_ mRNA levels were increased 7-fold (Fig. 1k). RNAscope for _Adamts18_ transcripts combined with IF for Sma showed the increased expression specifically in myoepithelial cells (Fig. 1l).

To test whether _Wnt4_ was furthermore required for _Adamts18_ expression, we engrafted contralateral cleared fat pads with _WT._

$EGFP^+$ and $Wnt4^{-/-}.EGFP^+$ mammary epithelia and harvested the transplanted glands on day 8.5 of pregnancy when $Wnt4$-dependent canonical Wnt signaling activity peaks[6]. Levels of $Wnt4$ expression in the mutant grafts were 1% of $WT$ levels and $Adamts18$ expression was reduced to 35% of $WT$ levels (Fig. 1m). Thus, increased canonical Wnt signaling induces $Adamts18$ expression and both $PR$ and $Wnt4$ are required for $Adamts18$ mRNA expression. This indicates that myoepithelial $Adamts18$ expression is downstream of the luminal PR/Wnt4 axis.

Potentially, our conclusion could be confounded by lineage differentiation and cell specification defects resulting from $PR$ and $Wnt4$ deletions. In light of the finding that both $PR^{-/-}$ and $Wnt4^{-/-}$ epithelial cells can differentiate into milk secreting alveolar cells[29,30], major cell specification defects are improbable. Nevertheless, we examined the possibility of a lineage differentiation defect by determining the ratio of luminal and myoepithelial cells in the two mutants. FACS analysis of lineage-depleted $WT$ and $PR^{-/-}$ mammary cells showed no significant difference in the two cell lineages (Supplementary Fig. 1a). As the $Wnt4^{-/-}$ mice die on embryonic day 13, we resorted to transplanting $WT.EGFP^+$ and $Wnt4^{-/-}.EGFP^+$ mammary epithelia derived for embryonic mammary buds[6] to contralateral fat pads and quantified the percentage of Sma+ epithelial cells by IF. The percentage of myoepithelial cells was decreased from 34% in the $WT$ to 26% in the $Wnt4^{-/-}$ epithelium (Supplementary Fig. 1b). To gain more insights into the lineage deregulation, we went on to compare FACS-sorted GFP+ luminal and myoepithelial cells from conditionally $Wnt4$-deleted ($MMTV::Cre^+.Wnt4^{fl/fl}.mT/mG$) and control ($MMTV::Cre^+.Wnt4^{wt/wt}.mT/mG$) epithelia by Affymetrix microarray analysis. The number of genes differentially expressed between the two genotypes was almost twice as high in the myoepithelial than in the luminal cell populations (Supplementary Fig. 1c–e). Hence, despite a lineage defect, there are major gene expression changes in the myoepithelium. Gene set enrichment analysis (GSEA) of the differentially expressed genes revealed that signatures reflecting the activity of the canonical Wnt signaling target, $Myc$, and the expression of its target genes were decreased in the $Wnt4^{-/-}$ myoepithelial but not luminal cells (Supplementary Fig. 1f). Together these findings are consistent with the model that Wnt4 secreted by luminal cells activates canonical Wnt signaling in the myoepithelial cells[6]. $Wnt4$ was the most significantly down-modulated gene in the luminal compartment (Supplementary Fig. 1d). While expression of $Cytokeratin$ 5 ($Krt5$) a gene typically enriched in myoepithelial cells, was increased in the $Wnt4^{-/-}$ luminal cells no cell type-related gene signatures were identified (Supplementary Fig. 1d). In the myoepithelial cell population, the secreted Wnt signaling inhibitor, $Wif1$, was the most significantly down-modulated gene suggesting the existence of a negative feedback loop in intraepithelial homeostasis (Supplementary Fig. 1e). The stem and progenitor cell markers, $Sox9$ and $Lgr5$, were decreased (Supplementary Fig. 1e). $Adamts18$ was also among the down modulated genes but failed to reach statistical significance (Supplementary Fig. 1e). GSEA revealed furthermore a decreased stem cell signature and an increase in Tgf-$\beta$ targets in the $Wnt4^{-/-}$ myoepithelial cells (Supplementary Fig. 1g). Reactome pathway analysis revealed a protein interactome centered around cell-cell junction and cell junction organization as well as cell-cell communication (Supplementary Fig. 1h). Taken together, while the deletion of $Wnt4$ results in a stem cell defect with some consequent cell lineage defect, the gene is expressed in the luminal compartment and its deletion affects transcription mostly in the myoepithelial compartment where $Adamts18$ is expressed.

**Mammary gland development in $Adamts18^{-/-}$ mice.** To assess the functional importance of $Adamts18$ in mammary gland development, we generated mice homozygous for an allele lacking exons 8 and 9, which encode the Zn-binding catalytic site[31] and analyzed their inguinal mammary glands at critical developmental stages by whole mount stereomicroscopy. In prepubertal, 14-day-old $WT$ and $Adamts18^{-/-}$ littermates, the ductal system was rudimentary and of similar size in both genotypes (Fig. 2a). Consistently, extent of fat pad filling (Fig. 2b) and the number of branching points were comparable in prepubertal, 14-day-old, $WT$ and $Adamts18^{-/-}$ littermates (Fig. 2c). In pubertal, 4–6-week-old, $WT$ females, milk ducts grew by characteristic dichotomous branching, extended beyond the subiliac lymph node, and had enlarged tips, terminal end buds (TEBs) characteristic of this stage (Fig. 2d). In the $Adamts18^{-/-}$ littermates, ducts barely reached the lymph node (Fig. 2d). The extent of fat pad filling was reduced by 50% (Fig. 2e), the number of branching points by 60% (Fig. 2f) and the number of TEBs by 40% compared to the $WT$ counterparts (Fig. 2g). In adult, 14-week-old, females, the milk ducts reached the edges of fat pads in both genotypes. In $WT$ females, ductal complexity was increased through side branching whereas ducts of the $Adamts18^{-/-}$ littermates were simple (Fig. 2h) and the number of branching points was 58% of $WT$ (Fig. 2i). Thus, $Adamts18$ is required for ductal development both during puberty and adulthood.

Histological examination of mammary glands from 6-week-old mice revealed structurally normal ducts with intact luminal and myoepithelial layers in both genotypes (Fig. 2j). To address whether the observed delay in ductal elongation was due to increased cell death and/or decreased cell proliferation, we stained sections from pubertal glands for cleaved-caspase 3 and phosphorylated histone H3 (pHH3). The proportion of cleaved caspase3+ cells did not differ significantly (Fig. 2k) but the pHH3-index in $Adamts18^{-/-}$ mammary epithelia was reduced to 64% of $WT$ levels (Fig. 2l, m). Thus, the delayed ductal elongation is due to decreased cell proliferation.

**$Adamts18$ function in the mammary epithelium.** $Adamts18^{-/-}$ pups show a transient growth delay[18], which may indirectly affect mammary gland development. In addition, subfertility associated with abnormalities in the female reproductive tract, such as dorsoventral vagina or imperforate vagina of $Adamts18^{-/-}$ females[18] precluded analysis of mammary gland development during pregnancy. To discern the epithelial-intrinsic role of $Adamts18$ in ductal growth at later developmental stages, we grafted mammary epithelium from $WT.EGFP^+$ and $Adamts18^{-/-}.EGFP^+$ females to contralateral inguinal glands of 3-week-old $WT$ female mice surgically divested of their endogenous epithelium. To unequivocally distinguish the engrafted epithelium from host epithelium that could have been inadvertently left behind during surgery, the donor cells constitutively expressed an enhanced green fluorescent protein ($EGFP$) under control of a chicken $\beta$-actin promoter[32]. Six weeks after engraftment, outgrowths derived from $WT$ donors filled the host fat pads whereas the contralateral $Adamts18^{-/-}$ epithelia failed to do so (Fig. 3a) and the branching points were decreased by 33% (Fig. 3b). Twelve weeks after engraftment, both $WT.EGFP^+$ and $Adamts18^{-/-}.EGFP^+$ outgrowths filled the host fat pads but side branching was decreased in $Adamts18^{-/-}.EGFP^+$ epithelial grafts (Fig. 3c). Flow cytometry of dissociated glands showed a 30% reduction in $EGFP^+$ cells (Fig. 3d) consistent with decreased cell proliferation resulting in lower epithelial cell numbers and delayed branching. Thus, the mammary branching phenotype in $Adamts18^{-/-}$ females is intrinsic to the mammary epithelium.

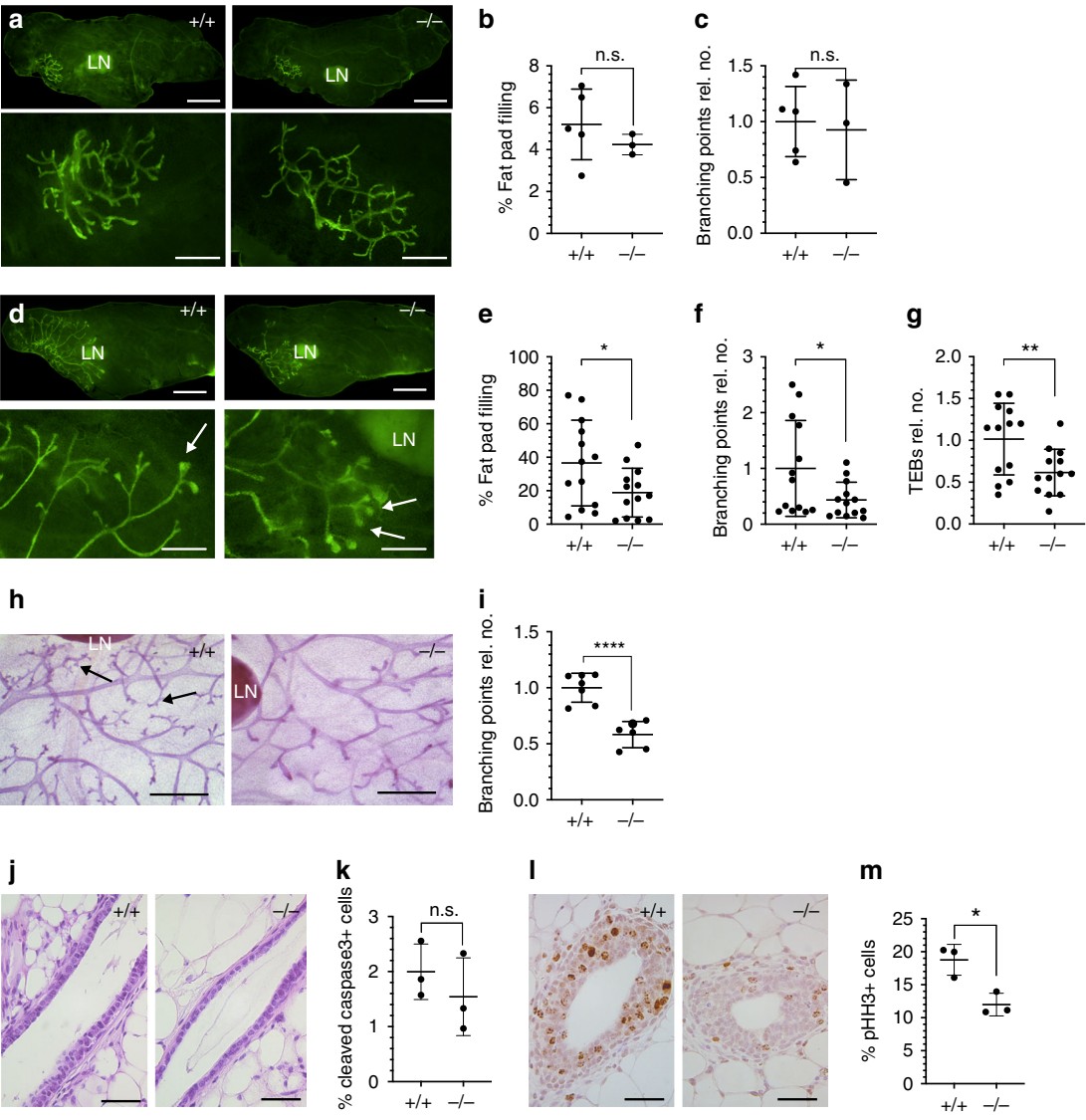

**Fig. 2 Mammary gland development in *Adamts18*−/− mice. a** Representative fluorescent stereo-micrographs of inguinal glands from 14-day-old prepubertal *WT.EGFP*+ and *Adamts18*−/−*.EGFP* females; *n* = 5 and *n* = 3. Scale bar, 500 μm. **b, c** Dot plots indicating the percentage of fat pad filling and fold change of branching points on day 14 mammary glands from 5 *WT.EGFP*+ and 3 *Adamts18*−/−*.EGFP* vs mice. Data represent mean ± SD unpaired Student *t*-test, two-tailed. **d** Representative fluorescent stereo-micrographs of inguinal glands from 4-week-old pubertal *WT.EGFP*+ and *Adamts18*−/−*.EGFP* mice; *n* = 13 for each genotype. LN: subiliac lymph node. Arrows indicate TEBs; scale bar, 500 μm. **e–g** Dot plots indicating percentage of fat pad filling, relative number of branching points and TEBs quantified at 4–6 weeks of age. Data represent mean ± SD from 13 *WT.EGFP*+ and 13 *Adamts18*−/−*.EGFP* mice. Unpaired Student *t*-test, two-tailed. **h** Representative stereo micrographs of whole mounted inguinal glands from 7 *WT* and 7 *Adamts18*−/− 14-week-old virgin mice. Arrows point to side branches; scale bars, 500 μm. **i** Dot plot showing relative number of branching points. Data represent mean ± SD from 7 *WT* and 7 *Adamts18*−/− 14-week-old virgin mice. Unpaired Student *t*-test, two-tailed. **j** Representative micrographs of H&E-stained histological sections of mammary glands from 5-week-old *WT* and *Adamts18*−/− littermates, *n* = 4; scale bar, 50 μm. **k** Dot plot showing percentage of cleaved caspase 3+ cells in TEBs of *WT* and *Adamts18*−/− females. Data represent mean ± SD from 3 *WT* and 3 *Adamts18*−/− mice. Unpaired Student *t*-test, two-tailed. **l** Representative pHH3 (brown) IHC on mammary glands from 6-week-old *WT* and *Adamts18*−/− females; hematoxylin counterstain; scale bar, 50 μm. **m** Dot plot showing the percentage of pHH3+ positive cells in TEBs of 3 *WT* and 3 *Adamts18*−/− females. Data represent mean ± SD, unpaired Student *t*-test, two-tailed. *$p < 0.05$; **$p < 0.01$; ***$p < 0.001$; ****$p < 0.0001$, n.s. not significant.

At 14.5 days of pregnancy, epithelia of both genotypes showed widespread alveoli both by fluorescence stereomicroscopy and histology (Fig. 3e). At day 1 of lactation, alveoli were fully distended (Fig. 3f) suggesting normal lactogenic function. However, at both time points, spaces between EGFP+ epithelial structures were larger in *Adamts18*−/−*.EGFP*+ grafts than in the *WT* counterparts consistent with reduced side branching at earlier stages (Fig. 3e–g). In line with the morphologic analysis and the decreased number of MECs, transcript levels of lactogenic differentiation markers such as *Lalba, Wap*, and *CsnA* were lower in mutant glands compared to

*WT* controls but failed to reach statistical significance when normalized to the epithelial marker *Krt18* (Fig. 3h). Thus, while epithelial cell numbers are decreased in the absence of *Adamts18*, the protease is not required for cytodifferentiation.

*Adamts18* expression has been reported in the stromal compartment and was confirmed by semi quantitative RT-PCR analysis of *WT* fat pads engrafted with *Adamts18*−/− epithelium showing 25% of the *Adamts18* transcript levels detected in *WT* recombinants (Supplementary Fig. 2a). To determine the functional importance of this stromal expression, *WT.EGFP*+

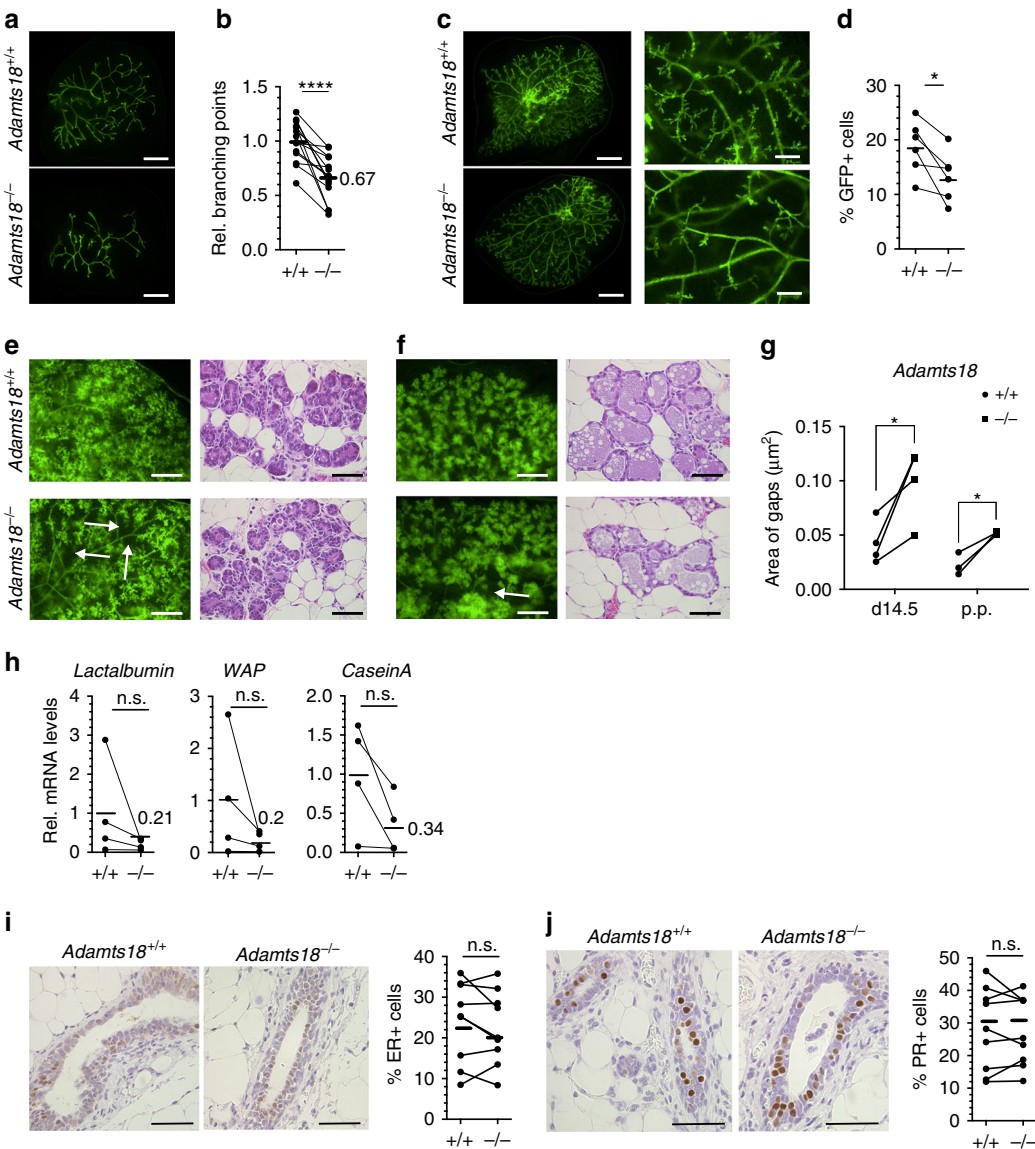

**Fig. 3 Mammary epithelial-intrinsic role of *Adamts18*. a** Fluorescence stereo-micrographs of contralateral glands 6 weeks after engraftment with *WT.EGFP+* or *Adamts18−/−.EGFP+* epithelia; scale bar, 500 μm. **b** Dot plot showing relative number of branching points in 14 contralateral *WT.EGFP+* and *Adamts18−/−. EGFP+* epithelial outgrowths; 4 different donors were used. **c** Representative fluorescence stereo micrographs of contralateral glands engrafted with *WT.EGFP+* or *Adamts18−/−.EGFP+* epithelium 12 weeks earlier. Scale bars, 500 μm (left), 150 μm (right). **d** Dot plot showing percentage of *GFP+* cells obtained from *WT. EGFP+* and *Adamts18−/−.EGFP+* contralateral grafts by flow cytometry. Tissue from 3 different donors was grafted to 6 pairs of contralateral fat pads. **e** Representative fluorescence stereo micrographs and micrographs of H&E-stained contralateral glands engrafted with *WT.EGFP+* (top) and *Adamts18−/−.EGFP+* (bottom) epithelia from host at day 14.5 of pregnancy. Three different donors were used, (*n* = 4). Arrows show interductal spaces; Scale bars, 5 mm and 50 μm. **f** Representative fluorescence stereo micrographs of 4 pairs of contralateral glands engrafted with *WT.EGFP+* and *Adamts18−/−.EGFP+* epithelia from host at lactation, 3 different donors were used in 3 independent experiments. H&E-stained micrographs thereof; scale bar, 50 μm. **g** Dot plots showing quantification of areas between branches from 4 pairs of contralateral glands at day 14.5 of pregnancy and 3 pairs of contralateral glands at post-partum. **h** Dot plots showing relative transcript levels of *Lactalbumin*, *Whey Acidic Protein (WAP)*, *Casein A* normalized to *Krt18*, in contralateral glands transplanted with *WT.EGFP+* and *Adamts18−/−.EGFP+* epithelia. Host is 14.5-day pregnant. Each pair of points represents an individual mouse; *n* = 4. **i** Representative micrographs showing IHC for ER on contralaterally engrafted *WT.EGFP+* and *Adamts18−/−.EGFP+* epithelia. Scale bar, 100 μm. Dot plot showing percentage of ER+ luminal cells in 9 contralateral grafts. **j** PR staining on contralateral glands engrafted with *WT.EGFP+* and *Adamts18−/−.EGFP+* epithelia. Scale bar, 50 μm. Dot plot showing percentage of PR+ luminal cells in 10 contralateral grafts. Statistical analysis by paired Student *t*-test, two-tailed. *$p < 0.05$; **$p < 0.01$; ***$p < 0.001$; ****$p < 0.0001$, n.s. not significant.

mammary epithelium was transplanted into cleared inguinal mammary fat pads of 3-week-old *Adamts18−/−* and *WT* mice. The recombined tissues were contralaterally transplanted unto the abdominal muscles of adult *WT* females. Fluorescence stereo-microscopy 6 weeks later showed that *WT* donor epithelium filled both *WT* and *Adamts18−/−* fat pads to comparable extent (Supplementary Fig. 2b, c). Twelve weeks after surgery, in both

*WT* and *Adamts18−/−* fat pads the implanted epithelia had developed side branches to comparable extent (Supplementary Fig. 2d). Thus, stromal *Adamts18* expression is not required for ductal branching.

As epithelial ER and PR signaling drive pubertal dichotomous branching and estrous cycle-induced side branching, respectively[29,33], we asked whether receptor expression was affected by

*Adamts18* inactivation. IHC of sections from contralateral glands engrafted with *WT.EGFP+* and *Adamts18−/−.EGFP+* epithelia revealed comparable proportions of ER+ (Fig. 3i) and PR+ cells (Fig. 3j) indicating that *Adamts18* is not required for ER or PR protein expression.

**The role of *Adamts18* in mammary epithelial self-renewal.** Delayed pubertal ductal outgrowth and reduced side branching together with normal alveologenesis and cytodifferentiation were previously observed in *Wnt4−/−* epithelia[26] and shown to result from a stem cell defect[6]. To test whether *Adamts18* deletion also affects mammary stem cells (MaSCs), we analyzed cells from dissociated *WT* and *Adamts18−/−* mammary glands by FACS[34] using CD24 and CD49f detection after depletion for lineage positive cells (Fig. 4a). The number of lineage-depleted cells obtained from mammary glands of 14-week-old females was one third less in *Adamts18−/−* compared to *WT* (Fig. 4b). The percentage of both luminal (Lin− CD24+ CD49f−) and myoepithelial (Lin− CD24^{low} CD49f^{low}) cells was not significantly altered in *Adamts18−/−* glands but the stromal cell fraction (Lin− CD24− CD49f−) increased by 19% in the mutant glands (Fig. 4c). Mammary progenitors, which give rise to colonies and are called colony forming cells (CFCs) represented <1% of the lineage negative cells in both genotypes whereas the number of MaSCs (Lin−CD24^{med}CD49f^{high}) also defined as mammary repopulating units (MRUs) was decreased by 43% in *Adamts18−/−* glands (Fig. 4a, d).

To functionally evaluate stem cell frequency in *WT* and *Adamts18−/−* mammary epithelia, we injected serially diluted single cells from *WT.EGFP+* and *Adamts18−/−.EGFP+* mammary glands to contralateral cleared fat pads of 3-week-old *WT.EGFP−* female mice. After 8 weeks, we determined frequency and extent of outgrowth by fluorescence stereomicroscopy combined with image analysis. The repopulating cell frequency[35] of *Adamts18−/−* cells was 10% of the *WT* cells with 1/20,000 vs. 1/2000 (Fig. 4e).

The single cell-based in vivo reconstitution assay can be confounded by cell adhesion and/or cell migration defects as well as by increased susceptibility to apoptosis. All these factors impact on any cell's ability, whether stem cell or not, to establish itself after injection in the fat pad, which is, of course, a prerequisite for the generation of any progeny. A complex assay that overcomes these limitations is the serial transplantation of pieces of intact epithelium. *WT* epithelium that is serially grafted can fill cleared fat pads for up to 7 generations[36]. Indeed, *WT. EGFP+* epithelium filled host fat pads efficiently over 5 transplant generations, however, the reconstitution ability of the contralaterally grafted *Adamts18−/−.EGFP+* epithelium decreased progressively to cease completely upon the 5th transplant (Fig. 4f, h, i). Histological analysis of the 4th generation transplants by H&E revealed no obvious difference (Fig. 4g). Thus, *Adamts18* is required for the regeneration capacity of the mammary epithelium, albeit to a lesser extent than *PR* and *Wnt4*, whose deletion blocks reconstitution at the 4th and 3rd generation, respectively, by the same assay[6].

**The basement membrane is part of the stem cell niche.** To address the mechanisms by which Adamts18 affects stem cell activity, we searched for its binding partners. In light of the myoepithelial cell-specific expression of the protease, we chose the human breast epithelial cell line, MCF10A, which has myoepithelial/basal characteristics[37], as a model. We ectopically expressed V5-tagged ADAMTS18 in these cells, immune precipitated it from the conditioned medium, and analyzed co-immunoprecipitated proteins by mass spectrometry. We discovered 238 proteins cumulatively in 3 independent experiments

(Supplementary Data 1), of which 31 were identified in ≥2 experiments (Fig. 5a). Transforming Growth Factor Beta-Induced (TGFBI), a secreted molecule that contains RGD domains similar to fibronectin and laminin and inhibits cellular adhesion to the ECM, was among the 12 proteins identified in all 3 experiments[38]. Bioinformatic analysis with MetaCore showed that top enriched MetaCore processes related to ECM organization and hemidesmosome assembly (Fig. 5b, Supplementary Table 1). The top localizations of the putative ADAMTS18 interactors were ECM, laminin-5 complex, and BM (Fig. 5c, Supplementary Table 2). Together, these findings support the hypothesis that Adamts18 function relates to the ECM and, more specifically, to the connection between epithelium and BM. This implies that the BM may be part of the stem cell niche.

To seek in vivo evidence for a role of the BM as part of the stem cell niche we turned to mice deficient for *Col18a1* because this heparin-sulfate proteoglycan is specifically localized to BMs[39]. Whole mount stereo-microscopy and morphometric analysis showed that *Col18a1−/−* females like their *Adamts18−/−* counterparts had delayed ductal elongation and fewer TEBs compared to their *WT* littermates (Fig. 5d). *Adamts18* and *Col18a1* double-deficient (*DKO*) mice showed a further decrease in TEB numbers, fat pad filling, and branching points at 6 weeks compared to single knockouts (Fig. 5e) indicating that *Adamts18* and *Col18a1* have additive roles in ductal elongation. To assess whether this genetic interaction affects stem cell function, we serially transplanted the *DKO* epithelium. While the contralateral *WT* epithelium reconstituted glands over 5 transplant cycles, the *DKO* epithelium failed to reconstitute by the 3rd generation (Fig. 5f–h). Thus, Adamts18 and Col18a1 cooperate in mammary stem cell control, providing in vivo evidence for a role of the BM in stem cell function, likely as part of the stem cell niche.

**Adamts18 modulates the ECM.** To probe for structural alterations in the ECM related to *Adamts18* deletion, we used picrosirius red to stain *Adamts18−/−* and *WT* pubertal mammary glands. Fibrillar collagen was increased around the ducts and TEBs in *Adamts18−/−* relative to *WT* (Fig. 6a). Immunoblotting of protein lysates from pubertal *WT* and *Adamts18−/−* glands and quantification showed that levels of the important BM components, laminin and collagen IV increased 1.7- and 3.9-fold, respectively, in *Adamts18−/−* glands (Fig. 6b, c). Levels of the major fibrillar collagen, collagen I, were increased 6.2-fold (Fig. 6b, c). Assembly of nascent collagen I, laminin and collagen IV matrices rely on initial assembly of fibrils composed of the primordial ECM glycoprotein fibronectin, the first ECM protein to be expressed during tissue development and wound healing[40,41]. Fibronectin levels were 3.2-fold higher in the mutants than in *WT* (Fig. 6b, c). IF showed increased staining intensity for all these proteins around ducts and TEBs in *Adamts18−/−* relative to *WT* pubertal glands (Fig. 6d). The staining was restricted to the BM for laminin and collagen IV but extended to the interstitial ECM for collagen I and fibronectin. Thus, in the absence of Adamts18, major ECM/BM components accumulate in the pubertal mammary gland in line with an important role for Adamts18 in ECM/BM remodeling.

Interestingly, analysis of mammary glands from 14-week-old *WT* and *Adamts18−/−* littermates showed that protein levels of laminin, collagens I and IV as well as fibronectin did not differ significantly between the two genotypes (Fig. 6e, f). This shows that Adamts18 is critical for ECM/BM modulation during pubertal ductal elongation and suggest that this specific developmental window determines mammary stem cell function.

**Adamts18 cleaves fibronectin.** In contrast with the increased fibronectin protein levels, its mRNA levels were unaltered in the

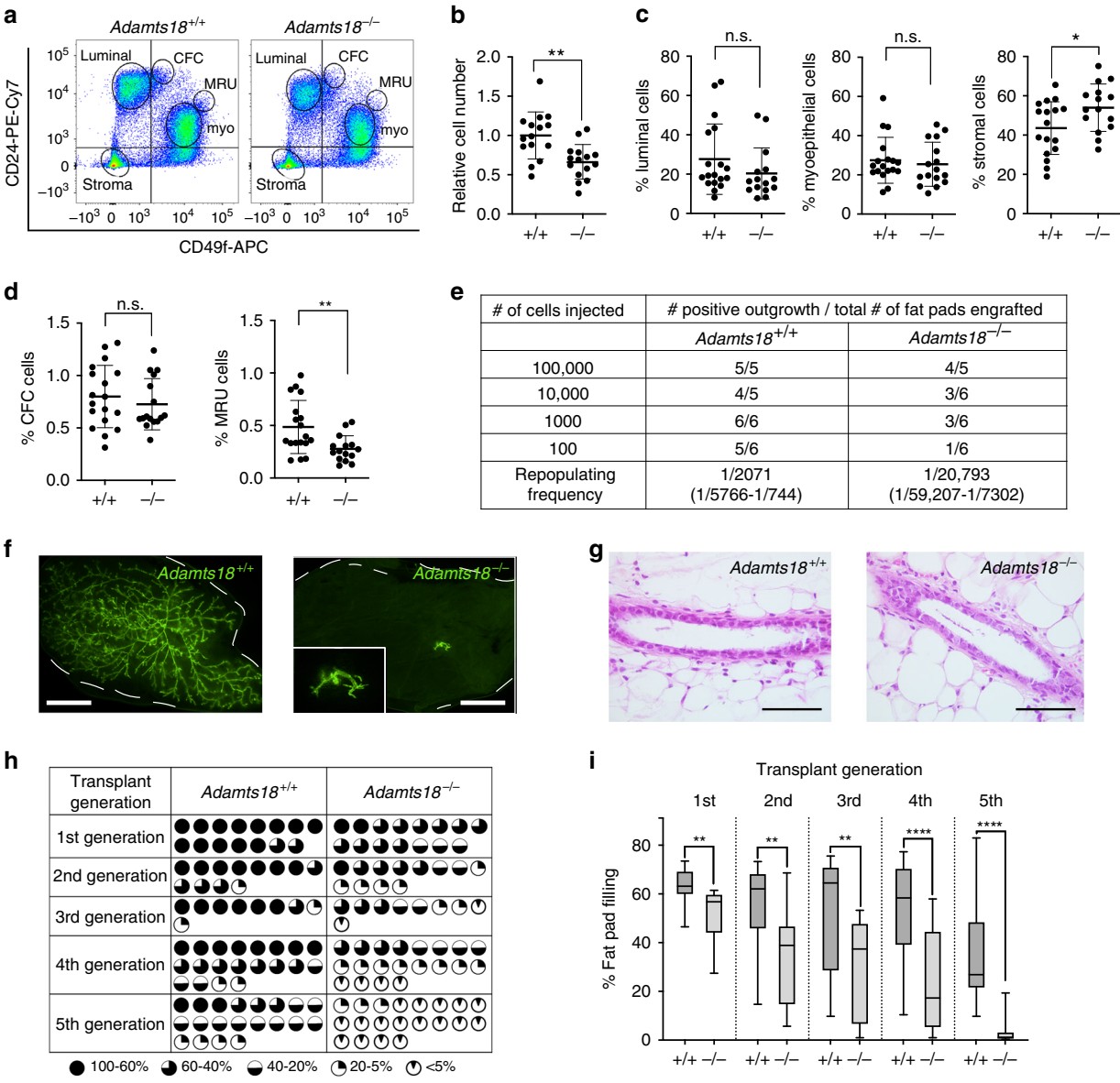

**Fig. 4 Role of *Adamts18* in the regenerative capacity of the mammary epithelium. a** Representative FACS dot plot showing CD49f and CD24 expression in the Lin⁻ mammary cells from 14-week-old *WT* and *Adamts18⁻/⁻* littermates. **b** Relative number of cells isolated from 15 14-week-old *WT* and *Adamts18⁻/⁻* littermates. **c, d** Dot plots showing lineage negative mammary cell populations from the mammary glands of 14-week-old *WT* and *Adamts18⁻/⁻* littermates. Data represent mean ± SD from 18 *WT* and 15 *Adamts18⁻/⁻* mice. Unpaired Student *t*-test, two-tailed. Total number of cells (**b**), percentage of luminal, myoepithelial, stromal cells (**c**), CFC and MRUs (**d**). Data represent mean ± SD from 18 *WT* and 15 *Adamts18⁻/⁻* mice. Unpaired Student *t*-test, two-tailed. **e** Table showing mammary outgrowths derived from *WT.EGFP⁺* and *Adamts18⁻/⁻.EGFP⁺* mammary cells injected at limiting dilutions into cleared contralateral fat pads. Positive outgrowths are defined as >5% fat pad area filled and related to total number of transplants. Repopulating cell frequency is shown, data are pooled from 3 independent experiments. **f** Fluorescence stereo-micrographs of contralateral mammary fat pads of recipient mice at the 5th generation of serial transplant after 8–12 weeks; scale bar, 500 μm. **g** Micrographs of H&E- stained histological sections of 4th generation transplants; scale bar, 50 μm. **h** Table summarizing 3 independent serial transplant experiments with *WT.EGFP⁺* and *Adamts18⁻/⁻.EGFP⁺* epithelia. Each engrafted gland is represented by a micrograph; black sectors represent area of fat pad filled by grafted epithelium. **i** Box plot showing extent of fat pad filling of contralateral grafts in each transplant generation. Boxes span the 25th to 75th percentile, whiskers 1.5 times the interquartile range. *p*-Values were determined by Wilcoxon test. \**p* < 0.05; \*\**p* < 0.01; \*\*\**p* < 0.001; \*\*\*\**p* < 0.0001, n.s. not significant.

pubertal *Adamts18⁻/⁻* mammary glands (Fig. 6g) suggesting that the observed increased staining could result from translational or posttranslational changes attributable to lack of *Adamts18*. As fibronectin is the prime component of nascent ECM fibers and a substrate of the *Adamts18* homolog *Adamts16*, we tested whether it is equally an Adamts18 substrate. We purified the secreted active form of ADAMTS18 from HEK-293T cells and incubated it with N-terminal 70 kDa fibronectin. The exogenous fibronectin fragment migrated slightly faster when co-incubated with EDTA

and was undetectable in the presence of ADAMTS18 after 24 h. When the digest was supplemented with EDTA, which chelates the bivalent metal ions required for ADAMTS activity, no change in fibronectin abundance was seen (Fig. 6h). Additionally, HEK-293T cells expressing *ADAMTS18* or a control vector were incubated without or with the 70 kDa recombinant fibronectin. By western blot, the medium of cells expressing ADAMTS18, but not the control vector, showed a readily detectable 30 kDa fibronectin fragment (Fig. 6i) similar to that detected after

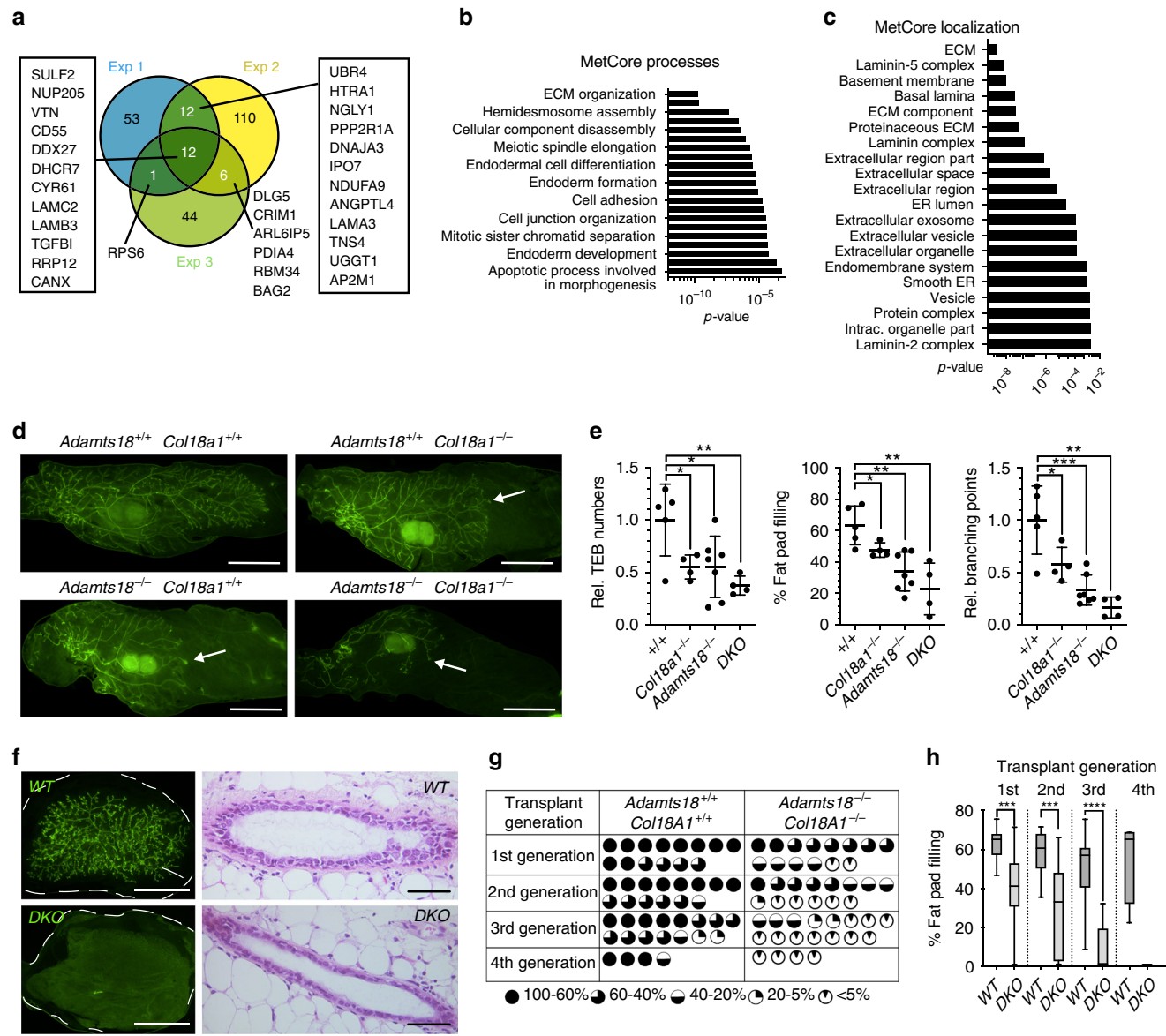

**Fig. 5 BM proteins and glycoproteins are involved in stem cell function. a** Venn diagram showing candidate ADAMTS18 binding proteins in the supernatant of MCF10A cells identified by affinity purification mass spectrometry. Gene names are shown in boxes for ease of recognition. Three independent experiments were done at different times. **b** Bar graph showing the top 20 METACORE processes of candidate ADAMTS18 binding partners based on $p$-values. **c** Bar graph showing top 20 METACORE localizations of candidate ADAMTS18 binding partners based on $p$-values. **d** Fluorescent stereo micrographs of representative 4th gland from $WT.EGFP^+$, $Col18a1^{-/-}.EGFP^+$, $Adamts18^{-/-}.EGFP^+$ and double-deficient $(DKO).EGFP^+$ 6-week-old pubertal mice; $n = 5$, 4, 7 and 4. Arrows indicate TEBs, scale bar, 500 μm. **e** Dot plots indicating the relative TEB numbers, percentage of fat pad filling, and relative branching points at 6 weeks of age. Data represent mean ± SD from 5 $WT$, 4 $Col18a1^{-/-}$, 7 $Adamts18^{-/-}$, and 4 $DKO$ mice. One-way analysis of variance (ANOVA). **f** Fluorescence stereo-micrographs of contralateral mammary fat pads of $Rag1^{-/-}$ recipient mice with 4th-generation mammary outgrowths derived from $WT.EGFP^+$ and $DKO.EGFP^+$ donor mice after 8–12 weeks; scale bar, 500 μm and H&E-stained sections thereof, scale bar 100 μm. **g** Table summarizing 3 independent serial transplant experiments with $WT.EGFP^+$ and $DKO.EGFP^+$ epithelia. Each engrafted gland is represented by a micrograph; black sectors represent the area of the fat pad filled by engrafted epithelium. **h** Box plot showing extent of fat pad filling of $WT.EGFP^+$ and $DKO.EGFP^+$ contralateral grafts in each transplant generation. Boxes span the 25th to 75th percentile, whiskers 1.5 times the interquartile range. $p$-Values were determined by Wilcoxon test. *$p < 0.05$; **$p < 0.01$; ***$p < 0.001$; ****$p < 0.0001$, n.s. not significant.

cleavage by $Adamts16$[21]. The amount of cleaved fibronectin increased 16-fold in the presence of ADAMTS18 (Fig. 6j). Thus, the presence of ADAMTS18 leads to fibronectin proteolysis, which may influence abundance of other ECM proteins and indirectly regulate growth factor availability and signaling.

**Stem cell signaling in $Adamts18^{-/-}$ glands.** Our findings pointed to the observed stem cell defect being secondary to changes in the ECM/BM. To elucidate the mechanisms by which

altered ECM affected stem cell signaling, we transcriptionally profiled 3 pairs of contralateral glands engrafted with either $WT$. $EGFP^+$ or $Adamts18^{-/-}.EGFP^+$ epithelia using RNA-seq. PCA analysis was used to identify and visualize possible batch effects due to sources of variation in the mice used (Supplementary Fig. 3a). After removing these effects by applying 2-way ANOVA correction, samples clustered by biological subgroups (Supplementary Fig. 3b). Expression of $Adamts18$, $Fgfr2$, and $Ctgf$ was tested and found reduced in all 3 $Adamts18^{-/-}$ samples after read

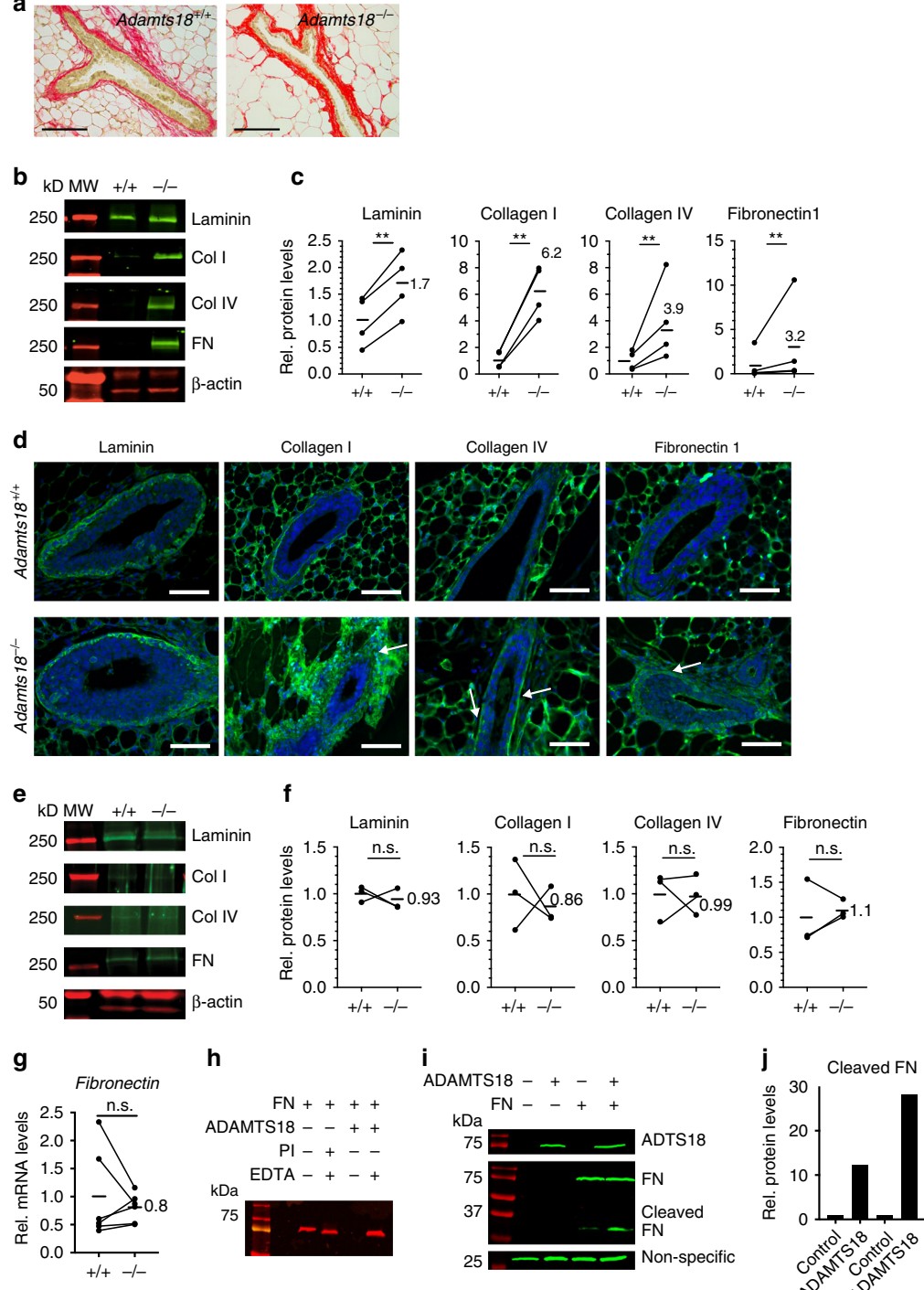

count normalization (Supplementary Fig. 3c). Overall, in *Adamts18*[−/−] transplanted glands, expression of 313 genes decreased (FC < 0.8, *p* < 0.05) and that of 273 genes increased (FC > 1.25, *p* < 0.05) (Fig. 7a). Analysis of the differentially expressed genes by pathway enrichment analysis using both ReactomePA[42] and ClusterProfiler[43] showed that cell junctions and ECM were affected, in particular various collagens and laminins (Supplementary Fig. 3d–g). More specifically, out of 40 significant GO terms, 11 were related to the ECM and 10 to Fgfr signaling, a pathway critical for stem cells[44,45] (Supplementary Table 3). Two of the 40 terms related to Hippo-Yap/Taz signaling another pathway critical for stem cell differentiation, which is upstream of *Fgfr2*[46]. When we specifically interrogated the genes whose

expression decreased, Reactome pathway analysis revealed Yap/Taz-mediated gene expression (Fig. 7b) and a protein interactome centered around cell-cell communication and cell-cell junctions as well as ECM, laminin and collagen complexes and assembly (Fig. 7c) that partly overlap with the Wnt4 specific interactome (Supplementary Fig. 1h).

In light of the increased ECM deposition, the differential expression of various ECM-related genes as well as the involvement of the Yap/Taz signaling pathway, we evaluated integrin expression in the *Adamts18*[−/−] glands. We extracted 27 *Integrin* genes, α and β Integrin subunits, from the RNAseq analysis and generated a heatmap (Supplementary Fig. 3h). No integrin-related gene was significantly altered by adjusted p-value,

**Fig. 6 Biochemical changes and Fibronectin cleavage elicited by Adamts18. a** Representative picrosirius red staining for fibrillar collagen (red) on 4th mammary gland sections from 5-week-old, pubertal *WT* and *Adamts18*$^{-/-}$ littermates; $n = 5$. Scale bar, 100 μm. **b** Representative western blot analysis on 3rd mammary glands of 5-week-old, pubertal *WT* and *Adamts18*$^{-/-}$ littermates; $n = 4$. β-actin loading control, MW marker in red. **c** Dot plots showing relative protein levels of laminin, collagen I, collagen IV, and fibronectin normalized to actin in 4 pubertal *WT* and *Adamts18*$^{-/-}$ littermates. Paired Student *t*-test, two-tailed; **$p < 0.01$. **d**, Fluorescent micrographs showing IF on 4th mammary gland sections from 5-week-old, pubertal *WT* and *Adamts18*$^{-/-}$ littermates for laminin, collagens I and IV as well as fibronectin (green) and DAPI nuclear stain (blue), $n = 3$. Arrows point to ECM density around TEBs or ducts; scale bar, 100 μm. **e** Representative western blot analysis on 3rd mammary glands of 14-week-old *WT* and *Adamts18*$^{-/-}$ littermates; $n = 3$. β-actin loading control, MW marker in red. **f** Dot plots showing relative protein levels of laminin, collagen I, collagen IV, and fibronectin normalized to actin in 3 adult *WT* and *Adamts18*$^{-/-}$ littermates. Paired Student *t*-test, two-tailed; n.s. not significant. **g** Dot plot showing relative transcript levels of *Fn1* normalized to *Hprt* in 3rd mammary glands from 6 pairs of 5-week-old *WT* and *Adamts18*$^{-/-}$ littermates. Paired Student *t*-test, two-tailed, n.s. not significant. **h** Representative Western blot analysis of 3 independent experiments in which fibronectin (FN)−70K was incubated with purified active Adamts18 in the presence or absence of EDTA and/or protease inhibitor (PI). Anti-FN antibody specific to the N-terminal heparin-binding domain. **i** Western blot analysis of FN1-70K incubated with ADAMTS18 overexpressing HEK-293T cells in the presence or absence of EDTA. **j** Bar graph showing levels of cleaved FN in supernatants from control transfected and *Adamts18* overexpressing HEK-293T cells in 2 independent experiments.

but *Itga3*, *Itgb4*, and *Itgb7* were significantly altered by *p*-value. Analysis of their expression levels by qRT-PCR at puberty in mammary glands from *WT* and *Adamts18*$^{-/-}$ mice showed *Itga3* and *Itgb4*, two integrins previously implicated in mammary stem cell function[47,48] and part of laminin 5 receptors, to be significantly down modulated in the mutants (Fig. 7d).

Together these findings suggest that Adamts18 is required for activation of the Hippo pathway, which in turn induces *Fgfr2* expression, activation of which is critical for stem cell function. Consistent with this scenario, the 3 Hippo target genes, *Ctgf*, *Fgfr2*, and *Gata3*[46,49] were reduced to 73%, 68% or 78% of *WT* levels, respectively, in additional transplants in the absence of *Adamts18* (Fig. 7e). Double-IF for Yap and the myoepithelial marker α-smooth muscle actin (Sma) showed expected nuclear localization of Yap in *WT* myoepithelial cells (Fig. 7f)[50]. In the contralateral *Adamts18*$^{-/-}$.*EGFP*$^+$ epithelia the signal intensity of Yap was decreased in myoepithelial cells (Fig. 7f). Quantitative image analysis revealed that the mean nuclear intensity of the Yap staining in the mutant epithelium was 58% of the contralateral *WT.EGFP*$^+$ transplanted glands (Fig. 7g).

To further support our claim that BM modulation by Adamts18 involves the Yap/Taz signaling pathway, we assessed the expression levels of downstream targets, *Cited-1*, *Ctgf*, *Fgfr2*, *Gata3* in pubertal *WT*, *Col18a1*$^{-/-}$, *Adamts18*$^{-/-}$, and *DKO* mice. Additionally, we assessed the expression levels of *Itga3* and *Itgb4* altered in *Adamts18*$^{-/-}$ mice. In line with our previous findings (Fig. 7h), we found the Yap/Taz targets to be significantly down modulated in pubertal *Adamts18*$^{-/-}$ and the *DKO*. *Col18a1*$^{-/-}$ glands displayed downmodulation in *Adamts18*, *Cited-1*, and *Ctgf*. This suggests that modulation of the BM composition by Adamts18 leads to activation of Yap/Taz signaling with increased *Fgfr2* expression and signaling which results in stem cell activation.

**ADAMTS18 in the human breast**. Our data indicate that Adamts18 translates the hormonal stimuli received by luminal cells into activation of stem cells via changes to the BM in the mouse mammary gland. To assess whether this signaling axis may also operate in the human breast, we generated a polyclonal antibody to ADAMTS18 and validated it on MCF10A over-expressing V5 tagged human ADAMTS18 with or without a short hairpin RNA (shRNA) to knock down overexpressed ADAMTS18 (Supplementary Fig. 4). IHC of reduction mammoplasty sections showed ADAMTS18 expression was not detected in the CK7+ luminal compartment, but in myoepithelial cells identified by p63 immunostaining (Fig. 8a) as observed for the transcripts in the mouse.

To test whether expression of *ADAMTS18* transcripts in human breast epithelial cells is similarly controlled by PR signaling, we humanized mouse mammary glands[51]. Human breast epithelial

cells isolated from 4 different reduction mammoplasty specimens were infected with lentiviruses expressing luciferase-GFP and injected into the milk ducts of immune-compromised *NOD scid gamma* females[51] (Fig. 8b). Once photon flux reached 10$^7$ per gland, the mice received subcutaneous pellets containing either vehicle, 20, or 50 mg progesterone (Fig. 8c, d). The hormone-containing pellets resulted in 7.2- and 19.7-fold increased plasma progesterone levels, respectively (Fig. 8c); *Adamts18* transcript levels were 1.8- and 2.3-fold higher than in noninjected mammary glands from the progesterone-treated mice, respectively, indicating that prolonged progesterone exposure results in increased *Adamts18* transcript levels in the mouse mammary glands (Fig. 8d). Next, we dissociated the xenografted glands to single cells and enriched for the human cells by depleting mouse cells with immunomagnetic beads. The xenografted cells from 4 different patients exposed to progesterone showed increased expression of *ADAMTS18* compared to control cells with an average 3-fold increase (Fig. 8e). Thus, the progesterone/ ADAMTS18 axis is conserved between mice and humans.

## Discussion
Here, we have addressed the longstanding puzzle of how epithelial ER and PR signaling connect to ECM changes that accompany both normal breast development and breast carcinogenesis. We show that the gene encoding Adamts18 is expressed in the myoepithelium downstream of Wnt4 secretion induced by ER/PR signaling luminal sensor cells (Fig. 9). The myoepithelial cells respond by canonical Wnt signaling activation and link luminal hormone receptor signaling to stromal changes with functional consequences. Our finding that altered BM composition affects MaSCs shows that the BM is a central part of the stem cell niche and a critical determinant of stem cell function.

The precise nature of the BM and interstitial ECM changes that alter signaling remain to be determined. Numerous factors, such as tissue stiffness and growth factor availability, directly or indirectly controlled by Adamts18 may be critical. The observed changes in the abundance of collagen I, collagen IV, laminin, fibronectin, and glycoproteins, like collagen XVIII, may be secondary to the reduced fibronectin clearance but Adamts18 may also be directly involved in their processing; other family members have glycoprotein substrates[14].

Increased laminin expression was also observed in *Adamts18*$^{-/-}$ adipose tissue[52] and embryonic brains[53] with effects on early adipocyte differentiation and spine and synapse formation. A detailed analysis of kidney and lung development in *WT* and *Adamts18*$^{-/-}$ mice revealed that expression of the enzyme by branching tips is important for branching and organ size[18].

We identified enhanced Yap/Taz nuclear localization and increased Fgfr2 signaling as potential mechanisms underlying

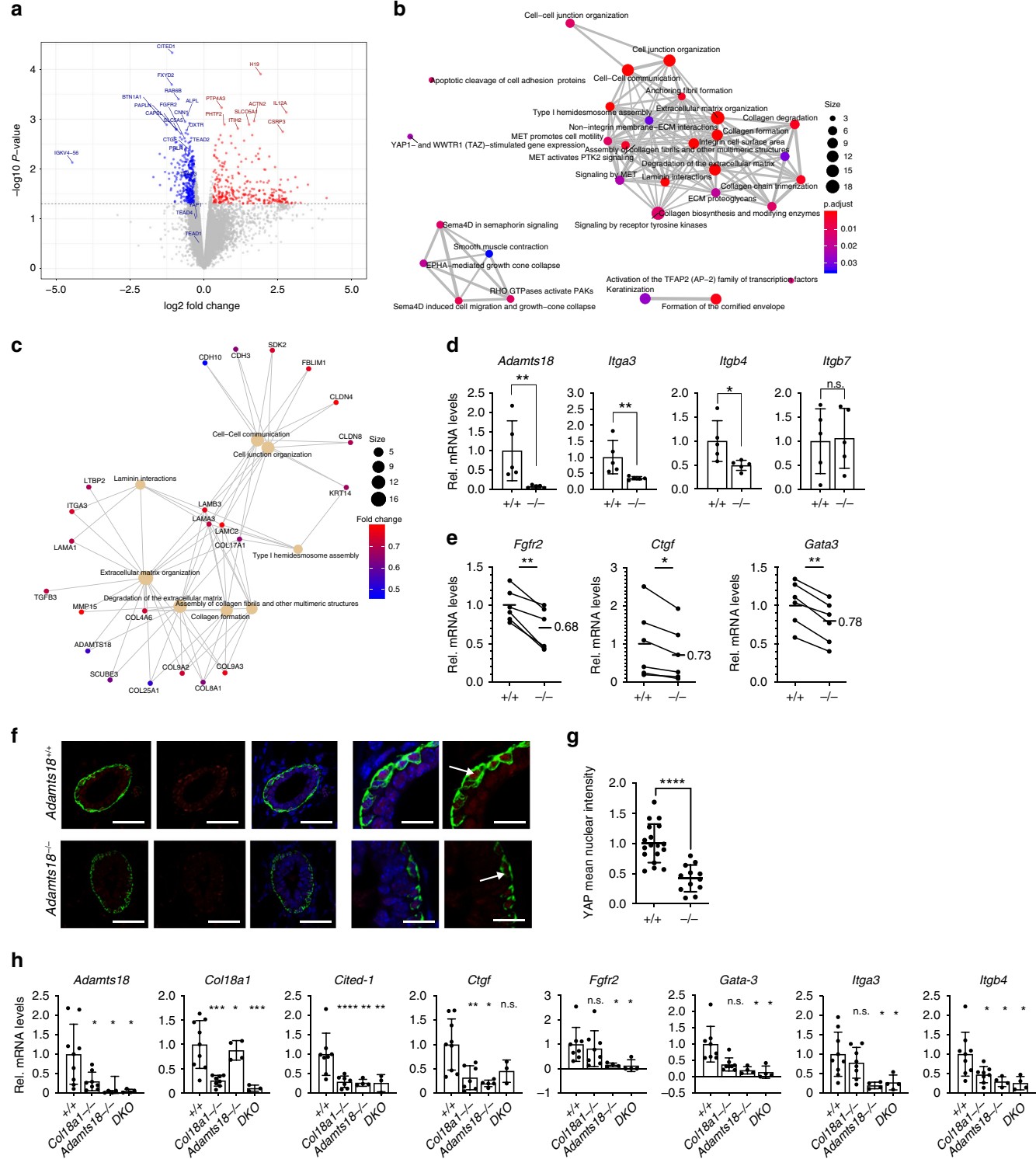

stem cell activation downstream of Adamts18 activity (Fig. 9). Whether Yap/Taz activation is central to increased Fgfr2 signaling and/or whether biochemical changes in the BM result in increased ligand availability was not addressed in our study. Yap/Taz signaling is typically activated by extracellular cues such as increased stiffness. Our gene expression analysis did not provide direct indications for this; whether the increased expression of muscle-related genes may also impinge on Yap/Taz or whether another stiffness independent mechanism[54] is important, remains to be explored. We speculate that Adamts18-induced modifications of

the ECM affect integrin-mediated, F-actin dependent cell-ECM adhesion and contraction, which promote cellular mechanical tension and Yap/Taz activation[55]. As such, the progesterone/Wnt4/Adamts18 axis provides an entry point for further studies of epithelial-BM interactions.

The regulatory axis we identified genetically in the mouse mammary gland likely operates in the human breast with implication for breast cancer prevention and treatment. Exposure to progesterone as it occurs recurrently during menstrual cycles has been shown to induce *WNT4* expression[56,57] and can increase

**Fig. 7 *Adamts18* impinges on transcription and regulates cell signaling. a** Volcano plot showing genes, which are differentially expressed between contralateral glands transplanted with *Adamts18*−/− and *WT* epithelia; n = 3, Kolmogorov–Smirnov test, all highlighted genes have *p*-values < 0.05. Genes with log2(FC) >0.5 in red and log2FC <0.5 in blue. Names of selected genes are indicated. **b** Enrichment map plot of Reactome pathway analysis (ReactomePA) on genes downregulated in 3 pairs of contralateral glands engrafted with *WT* and *Adamts18*−/− epithelia in 3 independent experiments with 3 different donors. **c** CNE plot of ReactomePA of genes down regulated in contralateral glands transplanted with *WT* and *Adamts18*−/− epithelia. **d** Bar graphs showing relative transcript levels of *Adamts18*, *Itga3*, *Itgb4*, and *Itgbt*, normalized to *Hprt* in 5 pubertal host mice bearing contralateral transplants of *WT* and *Adamts18*−/− epithelia. Data represent mean ± SD. Unpaired Student *t*-test, two-tailed. **e** Bar graphs showing relative transcript levels of *Fgfr2*, *Ctgf*, and *Gata3* normalized to *Hprt* in contralateral glands transplanted with *WT* and *Adamts18*−/− epithelia, n = 6. **f** Representative IF for Sma (green) and YAP (red) counterstained with DAPI (blue) of 4th mammary gland sections from 5-week-old *WT* and *Adamts18*−/− littermates; n = 3. Arrows indicate YAP positive nuclei of myoepithelial cells. **g** Dot plot showing quantification of relative mean intensity of nuclear YAP detected in myoepithelial cells of 5-week-old *WT* and *Adamts18*−/− littermates; n = 3. Each point represents an individual TEB. **h** Bar graphs showing relative transcript levels of *Adamts18*, *Col18a1*, *Cited-1*, *Ctgf*, *Fgfr2*, *Gata-3*, *Itga3*, and *Itgb4*, normalized to *Hprt* in pubertal *WT*, *Col18a1*−/−, *Adamts18*−/−, and *DKO*; n = 9, 8, 4, and 4, respectively. Data represent mean ± SD, one-way ANOVA. **p* < 0.05; ***p* < 0.01; ****p* < 0.001; *****p* < 0.0001, n.s. not significant.

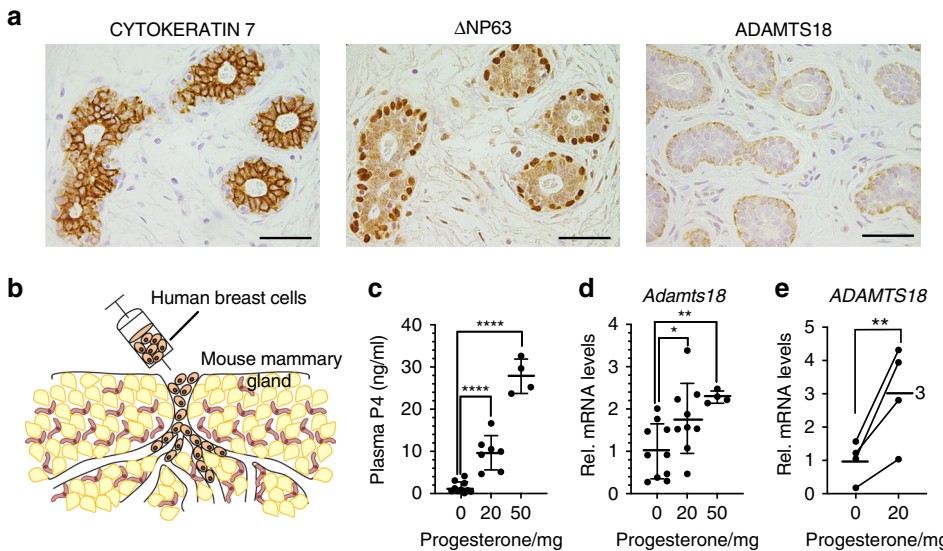

**Fig. 8 ADAMTS18 expression and distribution in the human breast epithelium. a** Representative micrographs of normal human breast tissue from 5 different women stained with luminal cell marker CK7, myoepithelial cell marker P63 and anti-ADAMTS18, counterstained with hematoxylin. Scale bar, 100 μm. **b** Experimental scheme: dissociated human breast epithelial cells from reduction mammoplasties were injected via the teat into the milk duct system of NSG female mice and establish themselves there. **c** LC/MS measured serum progesterone levels in mice 60 days after implantation with pellets containing vehicle, 20 or 50 mg progesterone. Data represent mean ± SD from n = 10 (vehicle), n = 7 (20 mg), and n = 4 (50 mg); one-way ANOVA. **d** Dot plot showing *Adamts18* transcript levels as measured by semi qRT-PCR normalized to the geometric mean of *Hprt* and *Gapdh* in mammary glands from mice that were subcutaneously engrafted with pellets containing either vehicle (0) or 20 or 50 mg progesterone for 60 days. Data represent mean ± SD from n = 10 (vehicle), n = 7 (20 mg), and n = 4 (50 mg); one-way ANOVA. **e** Dot plot showing relative *ADAMTS18* transcript levels normalized to *GAPDH* in glands xenografted with human breast epithelial cells from 4 mammoplasty specimens. Recipient mice were either implanted with vehicle- or 20 mg progesterone-containing pellets. Paired *t*-test, two-tailed. **p* < 0.05; ***p* < 0.01; ****p* < 0.001; *****p* < 0.0001.

ADAMTS18 expression, as we show here. The resulting BM/ECM remodeling may contribute to the increased breast cancer risk associated with recurrent menstrual cycles. Furthermore, the increased risk of postmenopausal women exposed to combined hormone replacement therapy with ethinyl estradiol and progestins may, at least in part, be attributable to increased stem cell divisions and stromal alterations[58,59] elicited by ADAMTS18.

Premenopausal patients with in situ carcinoma or early stage invasive disease, as well as women with high risk for breast cancer, may benefit from a preventive treatment that interferes with PR signaling or its downstream effectors. Blocking progesterone action, while possibly protective for the breast, will have many side effects as its actions are complex and affect many organs. Similarly, targeting downstream *Wnt* signaling has potential side effects because this signaling pathway is physiologically important for stem cells in many tissues. Based on the mouse model, ADAMTS18 is important for development of specific organs but it does not appear to have an essential

function in adult mice[31]. Furthermore, in its extracellular location ADAMTS18 makes it an excellent target for antibody-mediated therapy. As such, targeting ADAMTS18 appears as a feasible strategy for primary and secondary prevention unlikely to elicit major side effects.

## Methods

**Mice.** All mice were maintained and handled according to Swiss guidelines for animal safety and experiments were performed in accordance with protocols approved by the Service de la Consommation et des Affaires Vétérinaires of Canton de Vaud, Switzerland, with a 12-h-light-12-h-dark cycle, controlled temperature and food and water ad libitum. 129SV/C57BL6, mT/mG[60], and *NOD.Cg-Prkdcscid Il2rgtm1Wjl/SzJ (NSG)* mice were purchased from Jackson Laboratories and C57BL/6JOlaHsd mice from Harlan Laboratories. *Adamts18*−/− [31], *Col18a1*+/− [61], MMTV::Cre (lineA)[62], *Wnt4*+/− [63], *Wnt4*fl/fl [64], and *Tg(Act-EGFP)*[32] mice were maintained in C57BL/6JOlaHsd background.

**Patient sample processing.** The cantonal ethics committee approved the study (183/10). Breast tissue was obtained from women undergoing reduction mammoplasties with no previous history of breast cancer. All human subjects provided

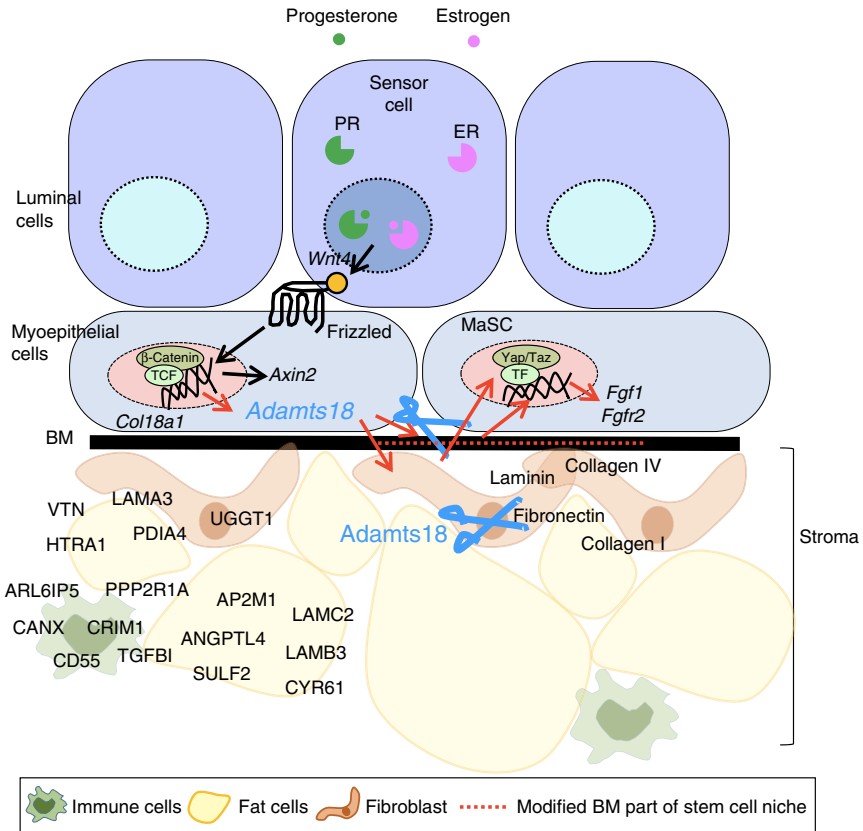

**Fig. 9 Working model of Adamts18 as a modulator of mammary gland development.** A schematic representation of the mammary acinar wall shows the spatial relationship between luminal cells, myoepithelial cells, BM and the surrounding interstitial ECM. Estrogen and progesterone induce Adamts18 production in myoepithelial cells via Wnt4-stimulated canonical Wnt signaling. Adamts18 remodels the BM and/or interstitial ECM, as part of the stem cell niche to ensure optimal stem cell regenerative capacity. Loss of *Adamts18* alters the stem cell niche and decreases mammary epithelial regenerative potential as its essential ECM modulatory function is abrogated.

informed consent for use of tissue samples in research. Samples were examined by the pathologist to be free of malignancy.

**Histology**. Inguinal mammary glands were fixed in 4% PFA in phosphate-buffered saline (PBS, pH 7.2) overnight at 4 °C, embedded in paraffin and cut into 4 μm sections. Hematoxylin and eosin or sirius red staining were performed according to standard protocols. For immunostaining, sections were de-waxed, rehydrated and subjected to antigen retrieval with 10 mM citrate buffer, pH 6.0 for 20 min at 95 °C. Sections were counterstained with Mayer's hematoxylin. For fluorescence microscopy, nuclei were counterstained with DAPI (Sigma). IF images were acquired on Leica DM 4000 B LED with Leica DFC 7000T camera and on Zeiss LSM700 confocal microscope for colocalizations. Primary antibodies A rabbit anti-ADAMTS18 antibody was raised against the peptide GQYKYPDKLPGQIYDA corresponding to ADAMTS18 sequence 502-516 aa (Eurogentec) an epitope that is conserved between human and mouse proteins, absent from other proteins and selected for high antigenic potential. The percentage of ER+ and PR+ cells were quantified using ImageJ, the percentage of SMA+ cells with QuPath software. Antibody list can be found in Supplementary Table 4.

**RNA in situ Hybridization**. *Adamts18* ISH was performed using RNAScope (Advanced Cell Diagnostics, Newark, CA) following the manufacturer's protocol. Briefly, 4 μm sections were deparaffinized and hybridized to a mouse *Adamts18* probe set (452251; Advanced Cell Diagnostics) using a HybEZ oven (Advanced Cell Diagnostics) and the RNAScope 2.5 HD Detection Reagent Kit (322360; Advanced Cell Diagnostics) and stained with anti-SMA after the RNAScope procedure.

**Transplantation**. Fat pads were transplanted onto the abdominal muscle wall of adult *WT* females[29]. Single cell suspensions of mammary epithelial cells in 20% matrigel were injected and 1 mm³ of epithelial fragments were transplanted to cleared fat pads. Intraductal injection of human breast epithelial cells was performed via cleaved teat.

**Mammary gland wholemounts**. Mammary gland whole-mounts were performed as described[65], and stereomicrographs were acquired with a LEICA MZ FLIII stereomicroscope and Leica MC170 HD. Fluorescence stereomicrographs were acquired on a LEICA M205FA with a Leica DFC 340FX camera. Fat pad filling and branching points were determined using ImageJ software.

**Single cell preparation**. Reduction mammoplasty microstructures were incubated with 1% collagenase A (Roche, final concentration of 1.0 mg/ml) in (DMEM)/F12 Dulbecco's modified Eagle's medium containing 1% penicillin/streptomycin (cat. 15070-063; Thermo Fisher Scientific) and 1% fungizone (cat. 15290-018; Thermo Fisher), overnight at 37 °C. Cells were dissociated to single cells with 0.25% trypsin-EDTA (Gibco, 15400), resuspended with red cell blood lysis buffer, and passed through 40 μm cell strainer. To isolate human cells from humanized mammary glands, single cells were incubated with mouse cell depletion cocktail (Miltenyi Biotec, 130-104-694) and passed through LS columns (130-042-401) on MACS separator according to manufacturer's protocol (Miltenyi Biotec).

**Hormone measurements**. Progesterone hormone levels in the plasma were measured using LC-MS (Q-Exactive, ThermoFisher Scientific)[66].

**Fluorescence activated cell sorting**. Single cell suspensions of mammary glands from 15- to 25-week-old virgin females were processed as described[34] and sorted on a FACSAria (Becton Dickinson).

**Hormone treatments**. Low consistency silicon elastomer (MED-4011) two parts (part A, MP3745/E81949 and part B, MP3744/E81950) were mixed with hormone powder, incubated at 37 °C overnight as described[67], and implanted subcutaneously. Three-week-old mice were ovariectomized and injected subcutaneously 10 days later with 17-β-estradiol 5 ng/g of body weight (Sigma–Aldrich, St. Louis, MO) using 5 mg/ml in 100% ethanol stock or vehicle. Mammary glands were harvested 18 h after injection.

**RT-PCR**. Mammary glands were homogenized with TRIzol reagent (Invitrogen), total RNA was isolated with miRNeasy Mini Kit (Qiagen), cDNA was synthesized with random p(dN)$_6$ primers (Roche) and MMLV reverse transcriptase (Invitrogen). Real-time PCR analysis in triplicates was performed with SYBR Green FastMix (Quanta) reaction mix. Primers used for RT-PCR, see Supplementary Table 5.

**Protein extraction and western blot**. Total proteins from the 3rd mammary glands of 5- and 14-week-old *WT* and *Adamts18*$^{-/-}$ littermate mice were extracted in Nonidet P-40 (NP-40) lysis buffer (2% NP-40, 80 mM NaCl, 100 mM Tris–HCl and 0.1% SDS) with a tissue disruptor on ice. 500 μl of buffer was used for 100 mg tissue and debris was removed by centrifugation. Transfected MCF-7 and MCF-10A were lysed with RIPA lysis buffer supplemented with protease inhibitors and protein concentration measured with a BCA kit (Pierce). Equal amounts of protein samples were subjected to SDS–PAGE on an 8% gel and electroblotted to PVDF membranes. Membranes were probed with fibronectin (Abcam ab2413), collagen IV (Abcam ab6586), collagen I (Abcam ab34710), laminin (Abcam ab30320), Lamin B1 (Abcam AB16048), ADAMTS18 (Eurogetech) and β-actin (Sigma mab1501) antibodies. IRDye conjugated secondary antibodies were detected with Odyssey CLx membrane scanner with Li-COR and band intensities quantified by ImageJ.

**AP-MS analysis for ADAMTS18 binding proteins**. MCF-10A cells were spin-infected with an *ADAMTS18* lentivirus containing a V5 tag or *LacZ* control virus. Cells were cultured to confluence in 10 cm dishes. Proteins were extracted with RIPA lysis buffer supplemented with protease inhibitors and protein concentration measured with a BCA kit (Pierce). ADAMTS18 was immunoprecipitated from 1 mg of protein using anti-V5 antibody conjugated agarose beads (Sigma A7345). The immune precipitates were subjected to SDS-PAGE, the gel was stained with colloidal Coomasie blue (Biorad), bands were excised and subjected to reduction/alkylation followed by tryptic digestion and LC-MS/MS proteomic analysis. Detected peptides were mapped against the human protein database, label-free protein quantification was performed and affinity lists were constructed in Scaffold 4 Proteomics Software using a minimum of 2 peptides to identify the proteins with a peptide false discovery rate (FDR) of 0.1% and protein FDR of 0.3%.

**Cloning**. ΔCT-*Adamts18*-867aa cDNAs were amplified from cDNA library prepared from eyes and fused to FLAG-tag and His$_6$-tag at N-terminus by PCR and cloned into *Nhe*I and *Hind*III restriction sites of pcDNA3.1/Hygromycin expression vector (Invitrogen). Plasmids were purified with HighPure midiprep kit (Invitrogen).

**Fibronectin cleavage**. 500 ng purified 70 K fibronectin (Sigma) were mixed with 50 ng of purified ΔCt-ADAMTS18 in digestion buffer (50 mM Tris-HCl, pH 7.5, 150 mM NaCl, 10 mM CaCl$_2$, 5 μM ZnCl$_2$), incubated 24 h at 37 °C in presence or absence of EDTA (25 mM) and PI (Pierce), and analyzed by WB with ABC antibody. 24 h after ΔCt-ADAMTS18 transfected 293 T cells were supplemented with 2 μg/ml purified 70 K fibronectin (Sigma) and heparin (100 μg/ml).

Bioinformatic analysis: For details of RNA-seq and microarray analyses see Supplementary Methods.

**Statistics**. Prism 6 software (GraphPad) used for statistical analyses and the statistical tests with their reported *p*-values are indicated in each figure.

**Reporting Summary**. Further information on research design is available in the Nature Research Reporting Summary linked to this article.

## Data availability

The authors declare that all data supporting the findings of this study are available within the article and its Supplementary Information files or from the corresponding author upon reasonable request. The datasets generated and analyzed during the current study have been deposited in the GEO database under the accession code: GSE145717, for microarray data and GSE145680 for RNA sequencing.

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

## Acknowledgements
We thank D. Buric and J. Dubail for advice and technical assistance, J. Dessimoz at the EPFL histology core facility, O. Burri and A. Seitz at the EPFL bioimaging and optics platform (BIOP), R. Guiet at the EPFL flow cytometry core facility (FCCF), and L. Tauzin and A. Mozes and, B. Mangeat at the EPFL gene expression core facility (GECF) for technical assistance. $Col18a1^{-/-}$ mice were kindly provided by Bjorn R. Olsen, Harvard Medical School. D.A. and M.S. received funding from the Swiss Cancer Ligue KFS-3701-08-2015, SNF 31003A_162550/1 Hormonal and cell signaling control of mammary gland morphogenesis: The role of Adamts18 in epithelial-basal membrane interactions that control the stem cell function downstream of progesterone receptor signaling, and SNF 31003A_141248 Hormonal and cell signaling control of mammary gland morphogenesis: ER/PR and Notch signaling interactions, P.A, C.C, C.L, R.J. SNF 31003A_162550/1, and R.R. from NCCR-Oncology 1A1 and SNF FN-MHV.

## Author contributions
Investigation: D.A., P.A., C.C., C.L., M.B., M.S., M.C., R.R., and T.M.; Bioinformatic analysis: R.J., G.A., and P.B.; Writing: D.A., P.A., S.A., and C.B; Funding acquisition: C.B.

## Competing interests
The authors declare no competing interests.
