## [Peer Review File · Nature Communications]

Reviewers' Comments:

Reviewer #1:

Remarks to the Author:

In this manuscript Ataca et al. evaluated the connections between progesterone signaling in the luminal cells and downstream stromal changes during normal breast development. The authors found that progesterone receptor (PgR) induced Wnt4 signaling by luminal epithelial cells stimulates Adamts18 production in the basal cells. The authors evaluated this novel role of Adamts18 in mammary gland development and find that Adamts18 can remodel the basement membrane (BM) contributing to mammary epithelial stemness. Overall the manuscript is a fairly comprehensive study with novel findings.

Major points to be addressed:

- 1) In Figure 1, the authors should co-stain the mammary gland sections (or the sequential sections) using a basal-specific marker in order to establish that the Adamts18 expression is specific to the basal/myoepithelial layer. In addition, besides mRNA levels, protein levels should also be analyzed if there is suitable antibody available. The authors do show IHC for human breast in Figure 8.
- 2) The authors should also evaluate the Wnt signaling pathway components in their PgR^{-/-} mammary grafts in Figure 1k, to better dissect the "order" of the PgR-Wnt-Adam18 axis.
- 3) The IHC image of Adamts18^{-/-} gland in Figure 3m seems to have a reduced number of PgR positive cells as compared to the WT, while the quantitation does not reveal a difference. How are the authors quantifying the percentage of PgR positive cells and was the image a representative one? The number of samples analyzed is low, it would be better to have more cases.
- 4) Would inducible expression of Adamts18 at a later stage of development rescue the regenerative capacity of the mammary epithelium in the Adamts18^{-/-} glands?
- 5) In Figure 5f, the fibronectin staining looks non-specific, as seen by positive staining inside the duct. The authors should be more rigorous in their assessment of background versus real staining.
- 6) Given the increased ECM deposition, the differential expression of various ECM-related genes as well as the involvement of the YAP/TAZ signaling pathway, the authors should evaluate integrin expression/activity in the Adamts18^{-/-} glands.
- 7) Given that Adamts18 is required during different stages of development, the authors should assess the ECM deposition at various stages as well to make a definite connection.
- 8) The conclusions drawn between BM modulation of Adamts18, YAP/TAZ and Fgfr signaling to stem cell activation need to be better supported experimentally.
- 9) Could the authors better explain the need for performing experiments with Col18a1^{-/-} mice?

Minor points

- 1) The CD24 positivity in the basal cells described on line 17 (page4), Figure 1a and the Figure1a legend is inconsistent.
- 2) Is the magnification consistent between Figure2 j and k?
- 3) Figure 7a and 7d are missing error bars and statistical analyses.
- 4) The description of Figure 8d in the text is confusing and needs to be better explained.
- 5) Grammatical errors throughout the manuscript needs to be rectified.

Reviewer #2:

Remarks to the Author:

The manuscript entitled "The secreted protease Adamts18 links mammary epithelial hormone action to extracellular matrix changes and stem cell niche function" describes how expression of Adamts18 influences the extracellular matrix and basement membrane of mammary glands, with a direct effect on branching morphogenesis and stem cell self-renewal. This is a very interesting manuscript that explores how signals across different mammary epithelial subtypes can direct mammary gland morphogenesis, and may also have an implication in breast cancer development. However, several pitfalls take away some of the excitement about the findings.

Most of the histological analysis lack proper quantification and/or cell-specific staining to support the notion of luminal x myoepithelial Adamts1 expression. Most of the gene expression quantification assays (qPCR and RNAseq) utilized mammary whole tissue, rather than FACS-isolated cells to support the idea of cell specific mechanisms.

Moreover, the authors utilized a series of mouse models to propose a mechanism of regulation to Adamts18, however such experiments rarely addressed the fact that lineage specification and cell differentiation could represent the basis for the observed results, rather than a block on signaling transduction.

Lastly, the initial observation that Adamts18 expression was important and restrict to myoepithelial population got lost when the authors abruptly switch in to studying the stromal compartment of mammary glands, especially given the results presented on Fig.1, that show low expression of Adamts18 in such compartment.

Nonetheless, in order for the manuscript to be accepted for publication, the authors must 1) provide proper figure labeling, 2) convincing histological images that confirm that Adamts18 is exclusive to myoepithelial cells, 3) luminal and myoepithelial quantification from outgrowths originated from all transgenic mice utilized, and cell specific gene expression analysis, and 4) a more convincing evidence that signals coming from luminal cells influence the expression of Adamts18 in myoepithelial cells.

Minor points:

1) Throughout the manuscript, the authors used basal and myoepithelial terminology to refer to same cell population. Basal defines a cellular compartment while myoepithelial refers to a cellular state. The authors should fix this inconsistency.

2) The authors mentioned that "PgR+ luminal cells, via Wnt4, activate canonical Wnt signaling in myoepithelial cells, which express Adamts18", and to access whether canonical Wnt signaling controls Adamts18 expression, the authors utilized MMTV-driven Wnt1 overexpressing transgenic mouse, which indicated increased Adamts18 expression in whole mammary gland RNA analysis. Although this analysis suggested a link between Wnt signaling and Adamts18 expression, it does not support that such gene expression regulation comes from a cellular communication (luminal to myoepithelial), neither rules out that MMTV driven Wnt overexpression is activating Adamts18 in other cell types (stromal cells). This is of special concern given that the mammary gland histology/ISH provided to support such claims do not convincingly show Adamts18 restricted to myoepithelial cells – thus isolation of cells utilizing flow cytometry would better support the myoepithelial-biased Adamts18 increase. Most importantly, co-culture of luminal cells overexpressing Wnt with Wt myoepithelial cells followed by imaging or gene expression analysis would be important to support the cellular orientation of signaling effects.

3) Please indicate that mice were matched for estrous cycles in all the histological analysis that quantified ductal morphogenesis, outgrowth and branching point, as distinct estrous cycle would have an impact on the results.

4) The quantification of stem cell self-renewal was properly done, however it lacked histological images to back up the data.

5) Adamts18^{-/-} outgrowths demonstrated spaces between EGFP⁺ epithelial structure during pregnancy day 14.5, however there is no quantification of such phenotype, which is needed to support the authors claims.

We thank both reviewers very much for their constructive comments that helped us improve the manuscript. We have added new data and substantially rewritten the manuscript including the figure legends and marked the changes in red. The responses to the reviewers' comments (black) are typed in blue.

Reviewers' comments:

Reviewer #1 (Remarks to the Author):

In this manuscript Ataca et al. evaluated the connections between progesterone signaling in the luminal cells and downstream stromal changes during normal breast development. The authors found that progesterone receptor (PgR) induced Wnt4 signaling by luminal epithelial cells stimulates *Adamts18* production in the basal cells. The authors evaluated this novel role of *Adamts18* in mammary gland development and find that *Adamts18* can remodel the basement membrane (BM) contributing to mammary epithelial stemness. Overall the manuscript is a fairly comprehensive study with novel findings.

We thank the reviewer for her/his constructive comments, interest in this manuscript and the appreciation of comprehensiveness and novelty of our study.

Major points to be addressed:

1) In Figure 1, the authors should co-stain the mammary gland sections (or the sequential sections) using a basal-specific marker in order to establish that the *Adamts18* expression is specific to the basal/myoepithelial layer. In addition, besides mRNA levels, protein levels should also be analyzed if there is suitable antibody available. The authors do show IHC for human breast in Figure 8.

Following the reviewer's comment, we have replaced the panels showing *in situ* hybridization for merely *Adamts18* in Figure 1 with panels showing double stainings: RNAscope for *Adamts18* followed by antibody staining for the myoepithelial-specific marker *Sma*. These data are presented in the New Figure 1 panels c-e and 1j and show that *Adamts18* is expressed in the myoepithelial cells. We recurred to using RNAscope because none of the commercial antibodies we tested worked for IHC. The antibody we produced ourselves works for IHC on human sections, as shown in Figure 8. However, it failed to work for mouse sections as shown in the IF staining below that we add for the reviewer's information.

Figure 1 : Testing of antibody raised against and *Adamts18* peptide on mouse mammary gland tissue sections.

2) The authors should also evaluate the Wnt signaling pathway components in their PgR^{-/-} mammary grafts in Figure 1k, to better dissect the “order” of the PgR-Wnt-Adam18 axis. To better dissect the “order” of the PgR-Wnt-Adam18 axis, we have now analyzed the expression of additional Wnt signaling related genes that are known to be expressed in the mammary gland, *Wnt 1, 4, 5a, 7b, 10a, 11 Lgr5, Lrp5, Lrp6* by RT-PCR in contra lateral glands engrafted with *WT* and *PR^{-/-}* epithelia. We show that only *Wnt4, Lgr5, and Lrp6* are differentially expressed in the **new Fig. 1j** and mention this in the text, page 5, lines 15-20. We have shown previously that *Wnt4* is expressed in *PR⁺* luminal cells and that its transcription as well as that of the canonical Wnt signaling target *Axin2* are both decreased in the *PR^{-/-}* epithelium (Rajaram et al 2016). These findings were corroborated by the use of an *axin2:LacZ* reporter mouse (Rajaram et al 2016). We have added a reference to this publication page 5 line 15.

3) The IHC image of *Adamts18^{-/-}* gland in Figure 3m seems to have a reduced number of PgR positive cells as compared to the *WT*, while the quantitation does not reveal a difference. How are the authors quantifying the percentage of PgR positive cells and was the image a representative one? The number of samples analyzed is low, it would be better to have more cases.

We have replaced Figure 3m showing a PR IHC, which gave the impression that there are fewer PR positive cells in the *Adamts18^{-/-}* compared to *WT* mammary epithelium. New **Figure 3j** now shows a more representative ducts. We have also increased the numbers of samples analyzed; instead of previously 4 and 5 we have now 7 and 9 independent samples for the ER and PR IHC, respectively, new **Figure 3j**.

4) Would inducible expression of *Adamts18* at a later stage of development rescue the regenerative capacity of the mammary epithelium in the *Adamts18^{-/-}* glands?

To address this interesting question, we have now extended the analysis of the ECM proteins to a timepoint in adulthood (week 14) and find that there is no significant difference any more. This shows that the ECM phenotype is transient, **Fig. 6e, f** p ll. 6-10. Hence, expression of *Adamts18* and the ensuing BM modifications are critical during a specific time window around puberty suggesting that there may be a specific developmental window for stem cell determination. This is in line with rodent experiments and the epidemiological observations in humans that irradiation during puberty increases breast cancer risk more than at later stages in life.

5) In Figure 5f, the fibronectin staining looks non-specific, as seen by positive staining inside the duct. The authors should be more rigorous in their assessment of background versus real staining.

We have replaced the panel Figure 5f showing a fibronectin staining with non-specific apical staining likely attributable to the presence of secretory material in the ductal lumen with new panel **Fig. 6d**, in which the apical border is clear of unspecific staining. We agree with the reviewer that caution is required in interpreting IF and would like to point out that the conclusions about differential protein expression are based on the western blots shown in **Fig. 6b** and that the IF data is merely confirmatory.

6) Given the increased ECM deposition, the differential expression of various ECM-related genes as well as the involvement of the YAP/TAZ signaling pathway, the authors should evaluate integrin expression/activity in the *Adamts18^{-/-}* glands.

We analyzed the expression of integrins, shown as a heatmap in **Supplementary Figure 3h**. The RNA sequencing reveals that 3 integrins are differentially expressed with $p < 0.05$ but not adjusted p value. For two of these, *Itga3* and *Itgb4*, differential expression is validated by semiquantitative RT-PCR shown in the new **Fig. 7d**.

7) Given that *Adamts18* is required during different stages of development, the authors should assess the ECM deposition at various stages as well to make a definite connection.

Following the reviewer's suggestion, we have now investigated ECM deposition in mammary glands from 14 week-old *Adamts18^{-/-}* and *WT* females. As shown in the new **Fig. 6e** and **f**, there is no difference at this stage. Thus the ECM phenotype is transient and suggests that stem cell determination occurs in a specific window during puberty.

8) The conclusions drawn between BM modulation of Adamts18, YAP/TAZ and Fgfr signaling to stem cell activation need to be better supported experimentally.

To be better supported experimentally, the conclusions between BM modulation of Adamts18, Yap/Taz and Fgfr signaling to stem cell activation, we have rewritten passages of the manuscript and provide now additional data on Yap/Taz signaling and Fgfr2 expression in the combined *Col18a1^{-/-}* & *Adamts18^{-/-}* (*DKO*) mice, new **Fig. 7h**, page12 lines 17-25.

9) Could the authors better explain the need for performing experiments with *Col18a1^{-/-}* mice?

We found by IP Mass Spec that ADAMTS18 in MCF10A cells *in vitro* interacts with several basement membrane proteins and hemi-desmosomal proteins. This suggested that Adamts18 main function may be in the basement membrane. As Adamts18 affects stem cells, the basement membrane in turn may be part of the stem cell niche. The experiments with *Col18a1^{-/-}* mice were performed to provide additional evidence for the BM having a role as part of the stem cell niche. *Col18a1* is a gene whose product is well established to be specifically localized and important to the basement membrane; hence we reasoned that the *Col18a1^{-/-}* phenotype provides additional support for the hypothesis that basal membrane is part of the stem cell niche, which is currently still poorly defined. We have rewritten parts of the manuscript to better explain this, pp 9f lines 19-11.

Minor points

1) The CD24 positivity in the basal cells described on line 17 (page4), Figure 1a and the Figure1a legend is inconsistent.

We have corrected the CD24 positivity in the myoepithelial cells both on page 4, line 15 and in the **Fig. 1a** legend to make it consistent with the labeling in **Fig. 1a**.

2) Is the magnification consistent between Figure2 j and k?

The magnification between Fig. 2 j and k now **Fig. 2h** is indeed consistent and both pictures were taken close to the iliac lymph node to ensure correct comparison. We have added arrows to highlight the side branches present in the *WT* glands.

3) Figure 7a and 7d are missing error bars and statistical analyses.

Fig. 7a now **Fig. 6g** statistical analysis was added, in previous Fig. 7d now **Fig. 6j** a single representative experiment is shown.

4) The description of Figure 8d in the text is confusing and needs to be better explained.

We edited **Fig. 8b, c, and d** and improved the description of **Fig. 8d** to better explain the represented results.

5) Grammatical errors throughout the manuscript needs to be rectified.

Grammatical errors throughout the manuscript were rectified. Two native English speakers re-read the manuscript.

Reviewer #2 (Remarks to the Author):

The manuscript entitled "The secreted protease Adamts18 links mammary epithelial hormone action to extracellular matrix changes and stem cell niche function" describes how expression of Adamts18 influences the extracellular matrix and basement membrane of mammary glands, with a direct effect on branching morphogenesis and stem cell self-renewal. This is a very interesting manuscript that explores how signals across different mammary epithelial subtypes

can direct mammary gland morphogenesis, and may also have an implication in breast cancer development. However, several pitfalls take away some of the excitement about the findings. We thank this reviewer for her/his constructive comments, interest in this manuscript and the appreciation of the potential clinical relevance of our findings.

Most of the histological analysis lack proper quantification and/or cell-specific staining to support the notion of luminal x myoepithelial *Adamts1* expression. Most of the gene expression quantification assays (qPCR and RNAseq) utilized mammary whole tissue, rather than FACS-isolated cells to support the idea of cell specific mechanisms.
See response to 1-4 below.

Moreover, the authors utilized a series of mouse models to propose a mechanism of regulation to *Adamts18*, however such experiments rarely addressed the fact that lineage specification and cell differentiation could represent the basis for the observed results, rather than a block on signaling transduction.

The reviewer points to the possibility that lineage specification and cell differentiation could represent the basis for the observed phenotypes, rather than a block on signaling transduction. We now consider and address this possibility at the end of the first section p5/6.

Lastly, the initial observation that *Adamts18* expression was important and restrict to myoepithelial population got lost when the authors abruptly switch in to studying the stromal compartment of mammary glands, especially given the results presented on Fig.1, that show low expression of *Adamts18* in such compartment.

Following this comment, we have relegated the data on the role of *Adamts18* expression in the stromal compartment of mammary glands, i.e. the fat pad transplantations to **Supplementary Figure 2b-d** not to disturb the logical, epithelial-focused flow as the reviewer suggested.

Nonetheless, in order for the manuscript to be accepted for publication, the authors must 1) provide proper figure labeling, 2) convincing histological images that confirm that *Adamts18* is exclusive to myoepithelial cells, 3) luminal and myoepithelial quantification from outgrowths originated from all transgenic mice utilized, and cell specific gene expression analysis, and 4) a more convincing evidence that signals coming from luminal cells influence the expression of *Adamts18* in myoepithelial cells.

1. We have reworked **Figures 1-8** as well as the **Supplementary Figures 1-4** and improved the figure labeling.

2. We now show by double labeling that *Adamts18* transcript expression in the mouse localizes to myoepithelial cells. This was achieved by RNAscope for *Adamts18* combined with IF for the myoepithelial-specific marker, Sma, and is shown in the new **Fig. 1c, d, e, and I**.

3. We quantified luminal and myoepithelial cells from *PR*^{-/-} and *PR*^{+/+} as well as *Wnt4*^{-/-} and *Wnt4*^{+/+} outgrowths shown in new **Supplementary Figure 1a and b**.

4. To provide more convincing evidence that signals coming from luminal cells influence the expression of *Adamts18* in myoepithelial cells we have added a reference to our previous work page 5 line 27. We have shown previously that *Wnt4* is expressed in PR+ luminal cells and that its transcription as well as that of the canonical Wnt signaling target *Axin2* are both decreased in the PR^{-/-} epithelium (Rajaram et al 2016). These findings were corroborated by the use of an *axin2:LacZ* reporter mouse (Rajaram et al 2016).

We added data on Wnt signaling components in the *PR*^{-/-} epithelium in **new Fig. 1j** and mention this in the text, page 5, lines 12-17. Most importantly, we added global gene expression analysis of purified luminal and myoepithelial cells either *Wnt4*^{-/-} or *Wnt4*^{+/+} new **Supplementary Figure 1c-e, page 6**.

Minor points:

1) Throughout the manuscript, the authors used basal and myoepithelial terminology to refer to same cell population. Basal defines a cellular compartment while myoepithelial refers to a cellular state. The authors should fix this inconsistency.

We have corrected the basal and myoepithelial terminology throughout the manuscript.

2) The authors mentioned that “PgR+ luminal cells, via Wnt4, activate canonical Wnt signaling in myoepithelial cells, which express Adamts18”, and to assess whether canonical Wnt signaling controls Adamts18 expression, the authors utilized MMTV-driven Wnt1 overexpressing transgenic mouse, which indicated increased Adamts18 expression in whole mammary gland RNA analysis. Although this analysis suggested a link between Wnt signaling and Adamts18 expression, it does not support that such gene expression regulation comes from a cellular communication (luminal to myoepithelial), neither rules out that MMTV driven Wnt overexpression is activating Adamts18 in other cell types (stromal cells). This is of special concern given that the mammary gland histology/ISH provided to support such claims do not convincingly show Adamts18 restricted to myoepithelial cells – thus isolation of cells utilizing flow cytometry would better support the myoepithelial-biased Adamts18 increase. Most importantly, co-culture of luminal cells overexpressing Wnt with Wt myoepithelial cells followed by imaging or gene expression analysis would be important to support the cellular orientation of signaling effects.

Part of our model, namely that PR signaling in luminal cells induces Wnt4 specifically in a subset of the PR+ luminal cells, which in turn activates canonical Wnt signaling in myoepithelial cells is based on our previous work (Rajaram et al EMBO 2016). We provided genetic evidence that Wnt4 is transcribed in PR+ luminal cells and that its transcription as well as that of the canonical Wnt signaling target *Axin2* are both decreased in the *PR*^{-/-} epithelium and used of an *Axin2:LacZ* reporter mouse to demonstrate *Axin2* transcription in the myoepithelium (Rajaram et al 2016). We have now added a reference to this publication p5, line 15.

Furthermore, we have now performed double labeling by RNAscope and Sma antibody staining and show that the increased *Adamts18* mRNA expression in *MMTV::Wnt-1* mammary glands localizes to Sma+ myoepithelial cells, shown in the new **Fig. 11**. The co-culture experiments the reviewer stipulates are not helpful as MMTV driven gene expression is lost upon in vitro culture (Rajaram R. unpublished results).

3) Please indicate that mice were matched for estrous cycles in all the histological analysis that quantified ductal morphogenesis, outgrowth and branching point, as distinct estrous cycle would have an impact on the results.

Pubertal ductal outgrowth as described in **Fig. 2a-g** occurs prior to the onset of estrous cycles. For the side branching quantification the experiments shown in **Fig. 2h-i** on intact mice, were repeated in the transplantation setting **Fig. 3a-j**, where any pair of mutant versus *WT* epithelium stems from the same mouse, hence it is in exactly the same endocrine milieu. The same applies to the various gene expression studies and IHCs performed on different mutant epithelial transplants that are always compared to the contralateral *WT* graft **Fig1i,j, Fig. 7d, e, g, h**.

4) The quantification of stem cell self-renewal was properly done, however it lacked histological images to back up the data.

We have now added histological images to the quantification of stem cell self-renewal data, see new **Fig. 4g** and **5f**.

5) Adamts18^{-/-} outgrowths demonstrated spaces between EGFP+ epithelial structure during pregnancy day 14.5, however there is no quantification of such phenotype, which is needed to support the authors claims.

We have now quantified spaces between EGFP+ epithelial structure during pregnancy day 14.5, and added a graph in new **Fig. 3g**

Reviewers' Comments:

Reviewer #1:

Remarks to the Author:

The authors have addressed each of the specific points of the reviewers by editing the text and performing additional experiments. The revised manuscript is significantly improved.

Reviewer #2:

Remarks to the Author:

A series of new analysis were added to the original manuscript, which reflected the care of the authors to properly address this reviewer comments, and increased the excitement about the findings. Overall the revision improved the authors' message and conclusions.

However, a few points still remain to be fixed prior to publication.

- 1) Majority of gland images (IF and histology) still lack proper label and scale bar indications on figures;
- 2) Figures still lack the number of mice utilized per experiment and per conditions.
- 3) GSEA plots can only be considered statistically significant if FDR is lower than 1 - this information should be included to GSEA plots

REVIEWERS' COMMENTS:

Reviewer #1 (Remarks to the Author):

The authors have addressed each of the specific points of the reviewers by editing the text and performing additional experiments. The revised manuscript is significantly improved.

--

Reviewer #2 (Remarks to the Author):

A series of new analysis were added to the original manuscript, which reflected the care of the authors to properly address this reviewer comments, and increased the excitement about the findings. Overall the revision improved the authors' message and conclusions.

However, a few points still remain to be fixed prior to publication.

1) Majority of gland images (IF and histology) still lack proper label and scale bar indications on figures;

We have added scale bars to all the panels of Figure 7f and added the scale bar value to Fig. 8a.

2) Figures still lack the number of mice utilized per experiment and per conditions.

We have spelled out the number of mice utilized per experiment and per conditions in the figure legends.

3) GSEA plots can only be considered statistically significant if FDR is lower than 1 - this information should be included to GSEA plots

We have enlarged GSEA plots and show the adj p-values: New Supplementary Figure 1.

Thank you very much for your consideration,

Best wishes,